# The Necessity and Goodness of Animals in Sijistānī's *Kashf Al-Maḥjūb*

Peter Adamson * and Hanif Amin Beidokhti *

Faculty of Philosophy, Philosophy of Science and Religious Studies, Lehrstuhl für Spätantike und Arabische Philosophie, Ludwig Maximilian University of Munich, D-80539 Munich, Germany
* Correspondence: peter.adamson@lrz.uni-muenchen.de (P.A.); hanif.aminbeidokhti@campus.lmu.de (H.A.B.)

**Abstract:** The Neoplatonic notion of "emanation" implies a required progression through hierarchical stages, originating from the highest principle (the One or God) and cascading down through a series of principles. While this process is deemed necessary, it is also inherently good, even "choiceworthy", aligning with the identification of the first principle with the Good. Plotinus, a prominent Neoplatonist, emphasizes the beauty and goodness of the sensible world, governed by divine providence. This perspective, transmitted through Arabic adaptations of Plotinus, influences Islamic philosophers too. This paper delves into the thought of the Ismāʿīlī philosopher Abū Yaʿqūb al-Sijistānī (d. after. 349/971), exploring the interplay of necessity and goodness in his cosmology, with a focus on non-human animals. Sijistānī's Persian Uncovering the Veiled provides a unique perspective on animals, presenting them as both necessary unfoldings of the universal intellect and inherently good beings with intrinsic value. The paper concludes with an appendix featuring an improved edition and English translation of relevant passages.

**Keywords:** Ismailism; Shiism; soul; animals; Neoplatonism; emanation; cosmology; Islam; Islamic philosophy

## 1. Introduction

Mention of the Neoplatonic concept of "emanation" brings to mind most readily the idea of a necessary progression through stages. Emanation begins from a highest principle (the One, or God), which gives rise to a chain of further principles. The metaphorical language of emanation is meant to stress the inevitability of this process. Just as a fountain cannot help but bubble over and a light must shed its rays, so each principle in the series must generate the next one until all possible entities have been produced. While all of this is perfectly correct, it leaves something out, namely that it is not only necessary but also good, and even "choiceworthy", that the hierarchical procession unfolds. It could hardly be otherwise, given that the first principle is also identified with the Good. Thus, as is well known, while Plotinus does indulge in quite a few disparaging remarks about the sensible world—the final stage of the emanation—he is also at pains to defend the beauty and goodness of that world, which is governed by divine providence.

A formidable illustration is provided by Plotinus' longest work, the so-called *Großschrift*, which was split up by his student and editor Porphyry into four treatises (III. 8, V. 8, V. 5, II. 9, which are 30–33 in the chronological order). Another example would be the treatise entitled by Porphyry *On Providence* and split up by him into two (III. 2–3, 47–48) (see on these works, e.g., [1–7]). Some of this material may have been unknown in the Islamic world, since we are not sure whether the first three of the six *Enneads* were available in Arabic. But the message would have gotten through nonetheless, and the Arabic Plotinus duly identifies the descent of soul into matter as being both necessary and chosen voluntarily. This is based on a passage at *Enneads* IV. 8 (6) 5, which says there is no conflict between necessity (*anangke*) and the voluntary (*hekousion*) (for the passage and further discussion, see [8], and

for freedom in Plotinus, see further, e.g., [9,10]). This made it easier for subsequent authors to reconcile a Neoplatonist metaphysics with the Islamic commitment to a voluntary God. In thinkers powerfully influenced by Neoplatonism, like al-Fārābī and Ibn Sīnā (Avicenna), the procession of things from God is seen as both "necessary (*wājib*)" and, because it is not compelled and is an expression of perfect goodness, a matter of "volition (*irāda*)".

In the following paper, we will consider a specific example of this pattern of thought, as it is found in the Ismāʿīlī philosopher Abū Yaʿqūb al-Sijistānī (d. after. 349 AH/971 CE). Previous research has emphasized the broadly Plotinian hierarchy of principles in Sijistānī and his elaboration of Neoplatonic negative theology (see especially [11]). Our own contribution will be an exploration the interplay of necessity and goodness in Sijistānī's cosmology, specifically with reference to the case of non-human animals. We choose this focus because animals are discussed numerous times, and in rather unusual ways, in his Persian language *Kashf al-Maḥjūb*, *Uncovering the Veiled*, hereafter *Kashf* (We have used the Persian edition in [12], which is cited by chapter and section number; and compared the following manuscripts: MS Minovi 2857, MS Adabiyāt 194jīm, MS Dāneshgāh-e Tehrān 8798, and MS Majles-e Showrā 11692 (fuller description of the manuscripts will be provided before the edited excerpts). We have also consulted the French translation in [13]. There is a translation of some sections into English by H. Landolt in [14] (pp. 83–129). It has received somewhat less attention than another work by Sijistānī, the Arabic *Kitāb al-Yanābīʿ* (*Book of Wellsprings*, hereafter *Yanābīʿ*), especially thanks to the translation and commentary of *The Book of the Wellsprings* (*Kitāb al-Yanābīʿ*) in [15]. Our contention will be that Sijistānī explains the existence of animals from two perspectives. On the one hand, they are necessary as automatic unfoldings of the content of the universal intellect. On the other hand they are good, both in the sense that they are beneficial from a human perspective and in the sense that they have an intrinsic good of their own. In an appendix to the paper, we offer an improved edition and English translation of the relevant passages (see Appendix A).

## 2. The *Kashf* and Its Author

*Kashf al-maḥjūb* (*Discovering/Unveiling the Veiled*) is the only Persian work attributed to Abū Yaʿqūb Sijzī/Segzī, or Sijistānī (d. after 971), nickname "Cottonseed" (in Persian: *panbeʰ-dāneʰ*, in Arabic: *khays(sh)afūj or ḥabb al-quṭn*) (for more on him, see [16–21] (pp. 415–416)). The authenticity of the Persian text has been challenged in modern scholarship. Instead, it has been suggested that the extant Persian text may have been translated from a lost, unwitnessed Arabic original (see Mīnovī's note to his manuscript of the *Kashf al-maḥjūb*, which he personally transcribed from the Taqavī *unicum*: MS Mīnovī 139/1, fol. 2, 21. June 1927. Cf. [22] (p. 22); [23] (p. 28); [18] (p. 190); [11] (pp. 20–22, pp. 164–5); [24] (p. 12)). It seems, however, that deniers of a Persian original have all followed S. M. Stern ([22]). Nafīsī and Walker offer no independent argument. They both, like Corbin, suggest Nāṣer-e Khosrow as the possible adaptor. Landolt has suggested that the anonymous commentator of the *Philosophical Poem* (*Qaṣīda*) of Abū l-Haytham Jorjānī is a more likely candidate. See [17] (pp. 74–82, at 75, 107, fn. 1).

Yet several early, mostly hostile authors confirm that Sijistānī wrote a treatise titled *Kashf al-maḥjūb*. Among these witnesses are the anti-Ismāʿīlī Zaydī polemicist Abū l-Qāsim Bustī (d. 420/1030), the polymath Abū Rayḥān Bīrūnī (d. ca. 440/1048), and the Ismāʿīlī philosopher, theologian, and missionary Nāṣer-e Khosrow (d. between 465/1072 and 471/1078). Bīrūnī, in his *Inquiries on India* (*Taḥqīq mā li-l-Hind*) completed in 421/1030, and Nāṣer-e Khosrow, in *The Traveler's Provision* (*Zād al-musāfir*), mentioned Sijistānī's scandalous endorsement of metempsychosis (*tanāsokh*) in the *Kashf al-maḥjūb* ([25] (p. 441); [26] (pp. 48–49); *Kashf al-maḥjūb*, V. 3, 59–61). Notoriously, Abū l-Qāsim Bustī criticized two theological statements of Sijistānī in *Kashf al-maḥjūb* that do not exist in our text. This has led scholars to the conclusion not only that Sijistānī did not pen the Persian text, but also that it does not reproduce the Arabic original in full ([22] (p. 22)). Herman Landolt, however, has proven the contrary. It was Bustī who reproduced Sijistānī's original text, perhaps in an intentionally distorted way so that he could condemn and refute him. More-

over, he might never had seen the original *Kashf* and might have confused it with another work of Sijistānī's such as *The Book of the Wellsprings* (*Kitāb al-Yanābīʿ*), especially given that he did not mention and criticize Sijistānī for endorsing metempsychosis, which is a heretical doctrine according to the majority of Muslim theologians ([17] (Introduction, pp. 81–82); [27]). So far, no manuscript or fragment of the purported original has been identified, and early references do not specify the language in which it would have been written. ʿAbbās Zaryāb Khoyī deems the arguments against the originality of the Persian extant text weak, tentatively suggesting that in the absence of any evidence or fragments of an Arabic original, there is insufficient reason to doubt that the *Kashf* was written in Persian ([28]). Besides the reasoning Stern offered in favor of the supposition of an original Arabic version based on Bustī's dubious reconstruction, the strongest argument is derived from a passage of the *Kashf* (VII. 1, §1). In this passage, the author refers to a certain Bā Yaʿqūb, who could be identified with the author of the *Kashf*, Abū Yaʿqūb. He relates a sentence from Bā Yaʿqūb in Arabic that is supposed to make a simple point regarding the Arabic morphology of the root *b-ʿ-th* (arousing, resurrection, or resuscitation) and three cognates derived from it: *inbiʿāth* (arising), *munbaʿith* (the one arising), and *mabʿūth* (the one made to arise, resuscitated, resurrected). It is noteworthy that the use of proper names (*ism ẓāhir*) instead of first-person pronouns, while striking, is one of the rhetorical and stylistic devices commonly found in medieval Persian and Arabic texts. The sentence reads *chonānk Bā Yaʿqūb gūyad al-inbiʿāth infiʿāl min al-baʿth wa-l-munbaʿith al-munfaʿil wa-l-mabʿūth al-mafʿūl* (*Kashf* VII. 1, §1.9–10). One finds, however, a similarly pedantic approach regarding the term *qiyāma* (resurrection) in *al-Risāla al-bāhira*. Cf. [29] (Vol. 1, pp. 633–634); [30] (§7).

Whether Sijistānī originally wrote the *Kashf* in Persian or it was translated from an original Arabic into Persian, there is no doubt that our Persian text dates back to the 5th/11th century (on this, see [31] vol 2, p. 52)). Furthermore, all topics associated with the work by later authors find a corresponding passage in our extant text. Thus, we can conclude that our Persian text is a faithful rendering of the original Arabic text, if the postulation of such an original is accepted ([28]; [17] (Introduction, p. 82); [27]).

The *Kashf* is deliberately structured, adhering to a coherent philosophical strategy that commences with the principle of unity, the First, and delves into Sijistānī's renowned radical apophatic theology (see Table A3). The exploration then progresses according to the Neoplatonist emanative order through intellect, soul, nature, and sublunary creation (see Tables A4–A7). The work then deals with Prophethood and the Imamate (see Table A8), and culminates in the topic of resurrection, elucidating the cyclical return of creation to the First (see Table A9). Between a prologue and an epilogue, the book comprises seven treatises or Discourses (*maqālat*), each consisting of seven issues (*jastār*) (see Tables A2–A9). Both of these elements are of symbolic significance. Ismāʿīlīs believe in seven Imams descending from the first Imam, Imam ʿAlī ibn Abī Ṭālib (d. 40/661), who was Prophet Muhammad's son in law and cousin, the youngest person to embrace Islam, and the second Muslim after the Prophet's wife, Khadīja bint Khuwaylid (d. 619CE, three years before the advent of the Islamic calendar). The number seven thus symbolizes Ismāʿīlī theology and holds major prominence in its cosmology and ontology. It is associated with divine creation, cosmic order, and philosophy of history. Besides the seven pillars of Ismāʿīlī Islam, seven is the number of the heavenly spheres, which also represent different stages of spiritual ascent. Each heaven is associated with a specific Imām, known as the "speaker" or *nāṭiq*. They are the bearers of divine knowledge and serve as guides for the seeker on the path of spiritual realization. The numerical symbolism extends to seven prophetic cycles, each of which includes seven prophets.

The *jastār* (Issue), which constitutes a *maqālat* (Discourse) (compare with Tables A3–A9), symbolizes the emanative process or flow (derived from *jastan*: to issue, to spring forward) and is taken to be the equivalent of the Arabic term *yanbūʿ* (wellspring), which Sijistānī has used in his *Kitāb al-Yanābīʿ* (*The Book of the Wellsprings*) (On this, we follow Ebrāhīmī Dīnānī, who relies an unnamed professor of Persian literature, and Landolt, in contrast to other

scholars who read the word as *jostār* (inquiry). [17] (Introduction, p. 77); [27]; [32] (p. 336)).
The seven discourses of the *Kashf* are as follows (see also Table A2):

I.      Discourse One: On Unity (*tawḥīd*)
II.     Discourse Two: On Recalling the First Creation [the Intellect]
III.    Discourse Three: On the Second Creation [the Soul]
IV.     Discourse Four: On the Third Creation, namely Nature
V.      Discourse Five: On the Fourth Creation [sublunary creation]
VI.     Discourse Six: On the Fifth Creation [Prophethood—Imamate]
VII.    Discourse Seven: On Mentioning the Sixth Creation [the Resurrection and return]

The focus of this article is mainly on the fourth creation (*Discourse Five*), but we refer to others Discourses too.

### 3. Animals in the Cosmic Order

As promised, we will be seeing that Sijistānī has some surprising things to say about animals. His way of classifying them among existents is, however, not one of them. At *Kashf* I. 1.3, he adopts a traditional, broadly Aristotelian division of substances into corporeal and incorporeal (or "spiritual": *rūḥ*), with the former being subdivided into living and inanimate: literally "growing" and "unliving" (*nāmī* vs. *mavāt*). The animate substances, finally, are either plants or animals (*ḥayavān*). One notable feature of this division, however, is the initial suggestion that there is an umbrella term, "thing (*chīz*)", which includes both substances and accidents, and which *could have* included literally all existents if we were willing to describe God as a "thing". Sijistānī backs away from this, however: he already argued at the very start of the work (*Kashf* I. 1) that God is not a thing, and the first phrase (*agar. . . būdī*) in our passage is to be taken counterfactually, as he says at *Kashf* II. 2.2 that God is also not a "substance". (On "thing" as an all-embracing term equivalent to "existent" in *kalām*, and as appropriated by Ibn Sīnā (Avicenna), see [33–35].)

In Sijistānī's Neoplatonic way of thinking, classes or genera are not only mental categories but principles found at the level of the intellect. At that level, a class like "substance" is a unity, which still embraces all possible substances within it. At *Kashf* II. 7.1, he refers to this as "absolute substance (*jawhar-e moṭlaq*)" and says that it is the job of soul to "unfold (*goshādan*)" this unity into a plurality. In this passage, we also get the further information that "animal" is subdivided into rational and non-rational, with the former division once again subdivided into mortal and immortal, that is, human and angel (see Figure 1).

This looks, at first glance, unproblematic and even quite standard, but in fact it is disconcerting. As we saw in *Kashf* I. 1.3 and has just been confirmed here in *Kashf* II. 7.1, "animal" is a subdivision of bodily substance as opposed to spiritual substance. One would have expected angels to appear on the spiritual side of this divide, not for them to be classified under corporeal substances, as rational animals alongside humans. One may suspect that Sijistānī has lost track of this in his zeal to affirm the traditional definition of human as "mortal, rational animal" which would list the last three steps in his division: he introduces angels at the last step solely to have a contrast class for the mortality of humans. However, his further remarks in the following section *Kashf* II. 7.2 seem to involve another inconsistency (see Figure 1). We were just told that animate substances are divided into plants and animals, with rationality and the lack thereof being a division within animals. But now, he says that the non-rational is subdivided into plants and animals. This is not quite as problematic as the point about angels, though. Non-rationality could, after all, belong to two different classes that appear at two different levels in the tree of divisions (Cf. *Kitāb Ithbāt al-nubuwwāt* [36] (II. 1, pp. 133–134)).

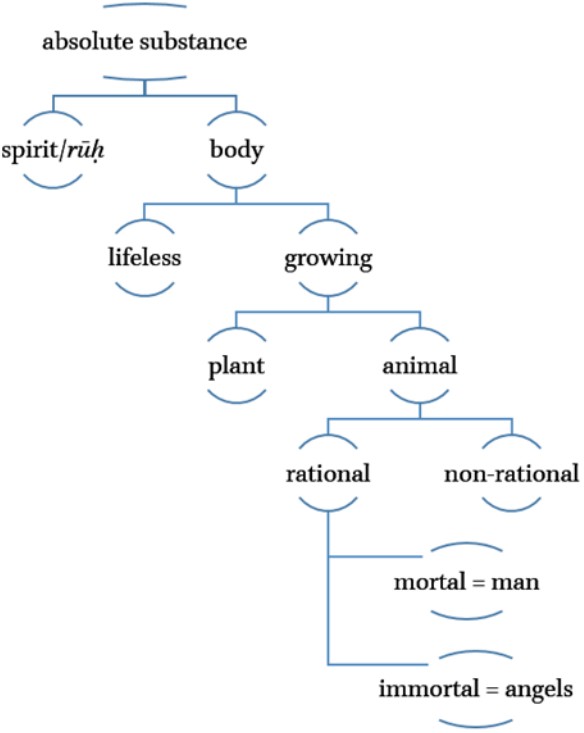

**Figure 1.** Unfolding, analysis, division (*Kashf* II. 7.1).

We should also attend to the different context of *Kashf* II. 7.2 (see Figure 2). In the first paragraph of *Kashf* II. 7, Sijistānī was describing the procession of substance from the level of intellect into the lower world through the ministrations of soul. Now, he seems to be talking about an epistemic process, whereby the soul "gathers together" what has been scattered through its own activity into different genera and species. So it may be that the soul does not just move through the same stages in backwards order (so that, in Neoplatonic terms, the "reversion" would perfectly track the steps of "procession"), but simply observes commonalities between classes and unites them together (see Figures 1 and 2). In any case, whether it is engaging in analysis ("bringing many from one substance") or synthesis ("bringing to one substance from many"), it can undertake the task only through conjoining (*joft gashtan*) itself to the intellect.

Humans have a special place in this scheme, because even as soul takes responsibility for dividing and disseminating absolute substance into its various types, it also "descends" and attaches especially to the human. This is explained at *Kashf* III. 1.1, where Sijistānī says that only humans have the needed "balance" or "harmony (*iʿtidāl*)" to accept the soul. Unfortunately, he does not expand upon this. The first thing we should address is the discrepancy between this passage and the evidence elsewhere in the text that for Sijistānī, plants and animals also are alive and have soul. (At *Kashf* II. 7.1 for example, he has already said that plants have a "growing soul": *nafs-e nāmī*.) One way to resolve this difficulty would be to say that only humans receive the *rational* soul or, what comes to the same thing, that only humans receive soul fully, with the whole range of its capacities—that is, including reason. In doing so, as we have already seen, humans come to have something in common with angels.

Next, we should ask what he means by the *iʿtidāl* that allows humans to receive soul. He may be referring to the "balance" of bodily humors, since it was a fundamental assumption of Galenic medicine that, among all animals, the human has the most perfectly balanced or "well-mixed" body. For instance, Galen wrote that the human is "in the middle, in its mixture, within the genus of animals as a whole" (see [37] (p. 107) citing Galen, *On Temperaments*, [38] (I. 541); [39] (20–21)). As Sijistānī goes on, though, he praises humankind as being "in the middle" in a second sense, namely that the human partakes of both

intellectual and sensible reality (here called "nature"). Humanity thus has a "border" on either side, which may recall Plotinus' metaphor that the soul is on the "horizon" between the two worlds (*Enneads* 4.4.3). In this same passage, Sijistānī also recalls the division of worldly substances into inanimate, plant, and animal. Previously, we were told that humans are a subdivision of animals, and that is borne out here when Sijistānī says that the human is distinguished by "sensible life (*zendagānī-ye ḥessī*)" as well as by intellective capacity. Life is even the "kernel (*lobb*)" or "center (*maghz*)" of nature. Thus, we may infer that the animate in general is the most central part of the sensible realm, while the human is the most outstanding representative of animate substance. We are, in short, the best of the best (*Kashf* VI. 1.1.)—though only on the level of species. On the level of individuals, beasts are superior to some human beings (*Kashf* V. 3.3.).

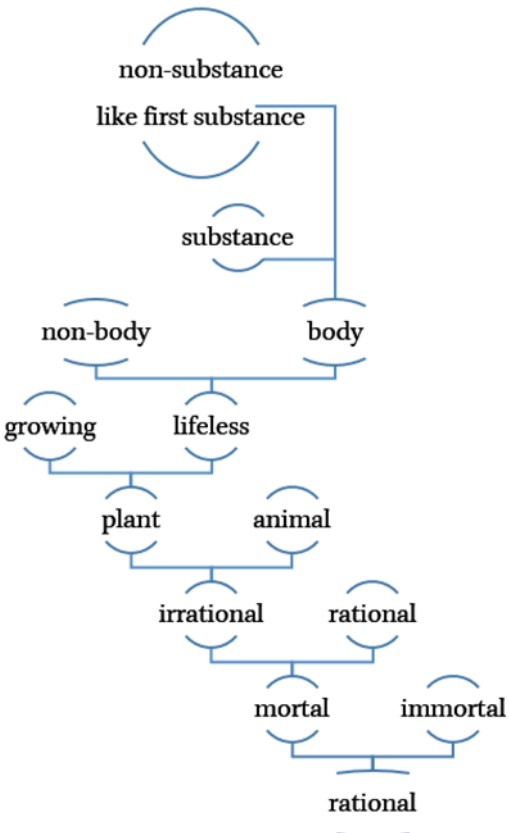

**Figure 2.** Reversion, synthesis, bringing together (*Kashf* II. 7.2).

So far, we have not seen Sijistānī say anything about the diversity of animal species, apart of course from humans. If humans are distinguished by the special descent of soul into our species, how are other animals distinguished from one another? This is a topic of some interest to Sijistānī, who devotes a chapter (*Kashf* V. 3) to the proposition that "species do not mix with one another, neither when they are composed nor after they have been composed (*naʰ-dar tarkīb wa naʰ-pas az tarkīb*)". His rationale for this in *Kashf* V. 3.1 gives us a first example of his pervasive attempt to explain cosmology by appealing to both necessity and goodness. He argues that if species could mix, then we would get individuals that partake of more than one species at a time, like a "half bird, half donkey". Note here his implicit appeal to something like the principle of plenitude: if mixing of species were even possible, *jāʾiz*, then it would in fact happen in particular individuals. But in fact, "such a thing does not exist and is impossible (*īn naʰ-mawjūd ast wa momtaneʿ ast*)".

The reason given for the first claim, that mixing does not exist, is that the "benefits (*manāfeʿ*)" of each species would be undermined (*bāṭel mī-shavad*). Again, a principle is being implicitly invoked here, in this case that the nature of each species is linked to certain

beneficial attributes or capacities. For example, it is beneficial for birds that they can fly and eat berries, neither of which is compatible with being a donkey. The supporting consideration for the second claim is harder to understand: "the mixing of species with one another is impossible because the species are connected to the individuals". We suggest understanding this as follows: as we have seen, what is unified at the level of intellect becomes variegated through soul's activity in the world of nature. In this process, more general classes are instantiated or manifested as *sub*-classes; for example, animals as a genus subordinated to the higher genus of animate substances. In the final step, an *infima species* will likewise manifest as individuals that fall under that species. For a single individual to fall under more than one animal species would violate this metaphysical law, so it is "impossible". An additional consideration, which Sijistānī immediately adds (*Kashf* V. 3.2), fits with this. Namely, that the individuals within a species have only the potentiality to generate something of the same species, so that a horse cannot generate a donkey or vice versa. One might think of this as the natural philosophical complement, or indeed consequence, of the metaphysical rationale just provided. A further consideration could be adduced, at the level of the intellect: if individuals of two species were to reproduce, the offspring would belong to a third species which is not represented at the level of the intelligible, and it is *this* that is impossible. Nevertheless, it should be noted that this is an awkward example for Sijistānī to have chosen, because the most famous and obvious example of the fact that animals of different species can indeed "mix" is the mule, which is the offspring of a horse and a donkey. Possibly, he would point to the infertility of mules as evidence that they are the exception that proves the rule. Alternatively, one could argue that the way he talks about the mixtures or half-man and half-donkey and half-bird half-donkey (*Kashf* V. 3.1) shows that he does not consider horse and donkey as separate species, since they share similar appearance and similar benefits. (On the relation between humans and horses in the Arabic zoological tradition, see [40]. On mules in Aristotle, and why mules, like all other mixed animals, are no species and have no forms, see [41]. On the classification of animals based on their appearance, see [42–45] (p. 12); [46,47].)

Sijistānī is still not done with this topic, though, because he wants to rule out another kind of "mixing", namely cross-species reincarnation, as when a human is born into the body of a dog or donkey or vice versa (for an account of various understandings of transmigration or metempsychosis, especially among Platonist philosophers, see [48]). He mounts an especially vigorous refutation against this proposal, perhaps because (as he says) he is aware of some ignorant people who actually accept the possibility of transmigration. They held that a human soul could enter the body of dog or donkey and vice versa. This is indeed a theory sometimes associated with Ismāʿīlism, and even with Sijistānī himself, on the authority of Bīrūnī and Nāṣer-e Khosrow (See further [11,49] (p. 99); [50,51]). The absence of that theory in our text is one factor that has made scholars suspect that the presumed translator of Sijistānī's *Kashf* has altered the original text.

Here, Sijistānī distances himself from transmigration (*tanāsokh*), calling it delirium (*hadhayān*), by building on the point just made, that individual members of species are generated from other members of the same species (*Kashf* V. 3.3). To refute the concept of cross-species transmigration, Sijistānī puts forth a theory of preformation concerning the embryo (*noṭfeh*). According to this theory, the form of doghood is invested and predestined (*maqdūr ast*) in the dog embryo or sperm. This embryo originates within the body of a male dog, passes into the body of a female dog, and is endowed with all the necessary attributes for sensation and movement, ultimately developing into a fully formed dog (*Kashf* V. 3.3) (cf. Aristotle. *Generation of Animals*, [52] (II. 3)). The celestial form of dog governs this whole process (*Kashf* V. 3.4), and there is no stage at which a human soul could attach to the dog's body. The possibility of cross-species transmigration would undermine the benefits and measures that are predestined in the species by God through the celestial forms that are the principles of the species. Moreover, cross-species transmigration would be pointless. The proponent of metamorphosis can find no theological and eschatological reasons for it because "human bodies are sufficient for the punishment of the sinners". Indeed, "there

are many in human bodies that are more corrupt and filthy than dogs, wolves, and swine!"
(*Kashf* V. 3.3). So it would be unjust for God to use animal bodies for punishing the sinners.
Hence, while Sijistānī denies the traditional notion of transmigration of souls, he makes
room for a restricted concept of transmigration, which is transmigration on the level of
the same species, such that the soul of a dog, after its death, could enter into the body of a
newborn dog, and the same for human souls (See [17] (Introduction, p. 80)).

Why would Sijistānī be so concerned with the preservation of species and with reject-
ing the possibility of their intermingling, while allowing inter-species reincarnation? We
can posit three theological and philosophical reasons. First, Sijistānī may be seeking to
establish that there is no infinity of souls, often raised as an absurd consequence against
the thesis of the eternity of the universe. Second and relatedly, by advocating for species
preservation, he ensures that there is a limit to individuals to be resurrected on the Final
Day (*Kashf* VII. 2. the relationship between the Final Day and reincarnation or rebirth is
stated in Sijistānī's *al-Risāla al-Bāhira* too; see [30] (§§14–15); [53] (pp. 198–218), and for
an English translation, [53] (pp. 60–75)). Third, Sijistānī considers the prophets to be a
distinct species, subordinate species to the human species, and may be inclined to confine
prophetic souls to a select group of bodies. This arrangement ensures that these sacred souls
reincarnate in various, perfectly balanced bodies. This aligns with the Ismāʿīlī philosophy
of history already presented above, which centers around cyclical recurrence.

Sijistānī wraps up his discussion of this "issue (*jastār*)" about the impossibility of
mixing species, by appealing to the celestial causes of animal species (*Kashf* V. 3.4). The
heavens provide the "principles (*aṣl-hā*)" from which the species "come forth (*padīd āyand*)".
Explaining this mechanism a bit more fully, he says, "there is for each species in creation a
celestial form, which preserves that species". (Sijistānī has a strikingly literal understanding
of the assignment of forms to the celestial spheres. In *Kashf* V. 7, one of his arguments
supporting the assertion that the number of species is fixed and determinate (*shemordeʰ*,
*maʿdūd*) is rooted in a distinctly corporeal interpretation of Platonic forms or species. He
writes, "the species are moulds (*qāleb-hā*) that are preserved in the round body (*jerm-e
modavvar*), namely in the sphere of the stars, such that they do not undergo any change
whatsoever (*keʰ az ḥāle- khʷīsh benagardand*). [...] Were the species unrestricted in number,
it would be permissible for them to become increased and diminished (*ziyādat-o noqṣān*).
Consequently, it would be necessary that increase and diminution appear in the spherical
body [the sphere of the stars]. The diminution of the spherical body would result in the
diminution of the species, and the diminution of the species result in the diminution of
the round body. It is, however, far from evident that a new form (*ṣūratī naw*) arises in
the round body a form residing in it (*ṣūratī az ṣūrat-hāye ū*) would be nullified" (*Kashf* V.
7.2, 3–11; Cf. *Kashf* IV. 3.3).) Yet again, the reader may be puzzled. Weren't we told that
the various classes of things in the natural world are produced by the soul unfolding the
unified content of intellect? How does this "celestial form", and heavenly causation more
generally, fit into that story? There are two chapters in the *Kashf* that bear on this question;
we will start with the more straightforward of the two, which is *Kashf* V. 6.

### 4. Animal Motion

Here, Sijistānī contends that the circular motion of the heavens is the cause for animal
motion. His argument to this effect in *Kashf* V. 6.1 runs as follows: whatever causes
animal motion must itself be a motion, since the cause should be in the same category as
the effect. Here, we mean "category" in the Aristotelian sense, since Sijistānī states that
motions cannot be caused by quantities, relations, or substances. Rather, the cause must
be "homogeneous (*moshākel*)", that is, of the same type as the effect. Of course, the idea
that every motion needs to be caused by another, prior motion is also solidly Aristotelian
(*Physics* VIII. 1). The next step once again displays Sijistānī's penchant for dichotomous
division: motion is either rectilinear or circular. But if the causing motion were rectilinear
then so would be the effect, and of course, animals do not always move in straight lines.
Thus, it is circular motion, which is found only in the heavens, that causes animal motion.

To make this argument go through, a number of ancillary assumptions or arguments are needed. For one thing, Sijistānī is clearly assuming that *complex* animal motions must have *simple* motions as their cause, since the contrast between rectilinear and circular is one made between simple motions. Even granting this, though, we would not have ruled out that there are multiple simple motions that collectively cause animal motion. A further problem is this: why wouldn't the same objection used against rectilinear motion also hold for circular motion? In other words, why doesn't the fact that animals don't move in circles show that they not (only) caused to move by circular motion? Sijistānī does give a response to this, in *Kashf* V. 6.2: animal motions have in common with celestial motions that there is not just one start- and end-point for them, whereas rectilinear motion (of an element, say) always goes from the same beginning to the same end (e.g., fire from "down" to "up"). Still, it has to be said that the whole line of argument is less than convincing. It looks very much like an attempt to justify on purely rational grounds something that is observed empirically, namely that animal behaviors like reproduction, sleeping and waking, and so on are keyed to celestial phenomena like the seasons and the difference between night and day.

The main contribution of *Kashf* V. 6 to our worry about the relation between celestial causation and the causation exercised by soul and intellect comes early in the chapter, before the whole line of argument just surveyed. It is something of a passing comment, to the effect that animal motions are a "branch" or "corollary" (*far<sup>c</sup>*) of nature, which "cannot fail to be (*chāre<sup>h</sup> nīst az ān*)". This suggests that animal motion is indeed integrated into the necessary, hierarchical production of all things from intellect through soul, since as we have seen the realm of nature is precisely the product of that procession. This suspicion is confirmed by the other passage we wanted to look at, which is *Kashf* V. 1, devoted to the "preservation of species". His discussion here applies patterns of argument we have already seen: in *Kashf* V. 1.1, he reasons from the need for genera and species to be manifested; in *Kashf* V. 1.2-3, he appeals to the influence of the celestial spheres.

In both cases, he emphasizes the *necessity* that animal species constantly persist. Without the species, the relevant individuals and genera would not survive. Here, one might suspect that he tacitly is thinking of these logical notions in mereological terms, that is, a genus is a whole made of species, while individuals are parts of a species. (He does not say this in so many words, but it could be inferred from his remark that the higher "gathers" what is below it: *gerd āvarad*; *Kashf* V. 1.1.) Since wholes can survive only if their parts survive, and vice versa, the vanishing of a species would eliminate both its individuals and its genus (for the Peripatetic background on metaphysics underlying the eternity of species, see [54]). Of course, this line of thought presupposes that, as Sijistānī himself says, "The persistence (*bemāndan*) of both the genera and the individuals is a necessity" (*Kashf* V. 1.1.). Unfortunately, it is not clear why this would be more obvious than the need for the species to persist. An opponent who thinks that a species could disappear would surely be happy for its individuals to disappear too, and might argue that the genus could survive so long as other of its species persist (when the dodo went extinct, the genus of birds continued to persist). Or, they might just allow for the possible disappearance of genera, since after all they are happy to allow the disappearance of species.

Since this was rather unsatisfying, let us turn to the second line of argument, about heavenly influence (*Kashf* V. 1.2). Sijistānī says that the species are "radiations (*sho<sup>c</sup>ā<sup>c</sup>āt*)" of the spheres, which stand at the "outmost ascending limit of nature (*nahāyat-e barshodan-e ṭabī<sup>c</sup>at*)" (cf. *Kitāb al-Yanābī<sup>c</sup>*, [15] (XIX, §§ 92–95)). Furthermore, there is no obstacle that could prevent the species from being radiated or received. This little passage solves a number of our worries about Sijistānī's view. First, it gives us a non-question-begging reason why species and their individuals are necessary: there is a cause that continually brings them about. Second, the "outermost" position of the heavenly realm may suggest that it is the transitional point, and causal intermediary, between the intelligible principles of intellect and soul on the one hand and the rest of the sublunary world on the other hand. Nature was often treated in the Neoplatonic tradition as a kind of fourth hypostasis (for instance by Plotinus in *Ennead* III. 8), and this conception is reflected in the layout of the

*Kashf*, whose first four major sections (*maqālāt*) deal in order with God, intellect, soul, *and nature*. Thus, we may understand the forms in the heavens as being the first reception, within the realm of nature, of forms that exist in a more unified way in intellect. Celestial causation then passes the forms on to things in the sublunary world, as we have seen with the specific case of astral motions causing animal motions (on the causal efficacy of the celestial spheres, see also *Kitāb al-Yanābīʿ*, [15] (§§ 63, 171, 186); *al-Risāla al-Bāhira*, [30] (§4)).

The whole line of argument might also put us in mind of the way the Active Intellect, the last celestial intellect, is the "giver of forms" in the cosmology of al-Fārābī and Ibn Sīnā. So it would be plausible to suppose that forms of the various species are disseminated into matter from the heavens. But the next paragraph (*Kashf* V. 1.3) contains a startling claim that at first glance looks incompatible with this idea. Sijistānī says that mineral, plant, and animal are not genera containing a multitude of species, but are instead each a *single* species. For example, "the multitude of the animals, like ants, gnats, horses, and camels, although they are all different in form (*beʰ-ṣūrat mokhtalef*), are all one species (*yek nawʿ ast*)". This looks like an overturning of Aristotelian biological class divisions, to say nothing of plain common sense. However, most likely, Sijistānī does not intend to deny that there are species below the level of the animal genus. He just wants to say that animality *as a whole* is passed on as a unity to, and by, the heavens. The difference between individuals and their types, within the large classes of mineral, plant, and animal, are due to "the difference in the mixture (*mezāj*) of the places" where the forms are received "by way of generation (*tavallod*)", as he says here speaking of minerals. Thus, he is able to go on immediately to distinguish between beneficial and non-beneficial "species" of animals, plants, and minerals (*Kashf* V. 1.4). The beneficial animal species include domesticated creatures like horses, cows, and sheep. On the non-beneficial side are beasts of prey (for the problem of harmful animals, see [55]; Sijistānī would fit nicely into Virgi's portrait of a tradition of texts that insist on placing such "unwanted" creatures within the providential order). Their species are also preserved because they too are consequences of the causal chain that comes down to the world of nature from intellect and soul through the heavens. As this passage shows, Sijistānī recognizes that necessity and benefit can come apart: harmful animals (as well as harmful plants and minerals) exist because they must, not because they do other creatures any good.

## 5. Animal Benefits

For the most part, though, plants and animals do plenty of good. In particular, they do good for humans. At *Kashf* III. 4, he takes up the question of what share (*bahr*) animals have in soul. He answers the question by using the terms "love (*maḥabbat*)" and "dominion (*ġalabeʰ*)" to designate, respectively, providential care and the dependency of that which receives such care (*Kashf* III. 4.1; for *maḥabbat* and *ġalabat* in relation to animal soul, see [Pseudo-]Aristotle, *Theology of Aristotle*: [56] (VIII, §166); [57] (pp. 98–99); [58] (p. 473); [59] (*Enn.* VI. 7.14)). Thus, intellect loves soul and sheds benefit upon it; soul is conversely dependent on intellect for the benefits it receives. The same dynamic is observable in nature: plants benefit animals by giving them nourishment; animals benefit humans by providing milk, meat, butter, wool, and fur. Sijistānī is, obviously enough, considering the goodness of animals from an anthropocentric perspective. The reason it is both good and necessary for many animal species to exist is that these animals benefit humans and facilitate human persistence.

Anthropocentrism also seems to be upheld at the start of the next paragraph (*Kashf* III. 4.2), where he seemingly ranks those animal species higher that can accept training and perform useful or human-like behaviors like hunting and talking (this is reminiscent of Miskawayh's hierarchy of animal species, which ranks them in terms of their ability to imitate or be trained by humans. See [60] (p. 13); Miskawayh's recommendation to avoid "injustice" towards animals is similar to the following remark of Sijistānī, in that both seem to be motivated by a belief in animal teleology). These animals are distinguished by their ability to "learn human *adab* (*hamī āmūzand adab-e mardom*)"; here, it is not animal

benefit for humans that is in focus, but Sijistānī is still evaluating them in terms of their similarity to humans. Yet in the immediate sequel, he shows himself able to appreciate that animals have a value in their own right: "There is still another share of the soul, nobler than these [aforementioned] shares, in the animals, which is their longing for persistence (*baqāʾ*) and their fleeing from destruction (*fanāʾ*), because there is no animal that does not by its own nature fear death and strive to stay alive. It is for this reason that one does not kill an animal unless there is a benefit [in it]". This may be the most noteworthy passage on animals in the whole *Kashf*, showing as it does that for Sijistānī, animals themselves benefit from their own existence. Certainly, he thinks that this value is not absolute: even as he tells us that we shouldn't kill animals for no good reason, he allows us to do it when we do have good reason, like preservation of another species. Presumably the "benefit" to be had from, say, eating meat would for him count as a good enough reason: Sijistānī was no animal liberationist. Still, his acknowledgement of a non-anthropocentric teleology in animal life leads him to give at least some moral weight to the interests of animals, which in this historical context is noteworthy. Ironically, it puts him in agreement with an earlier near contemporary who was a bitter enemy of the Ismāʿīlīs, namely Abū Bakr al-Rāzī (See [61]).

Respect for animals is also indicated by another section, which takes up the curious question of whether plants are prior (*pīsh būdī*) to animals, followed by humans (*Kashf* V. 2.1). Apparently, the idea of Sijistānī's unidentified opponents, who most probably are the Brethren of Purity (*Ikhwān al-Ṣafāʾ*) in the *Epistles* XXI–XXII on the genera of plants and the generation of animals (cf. [62], vol. 2), is that more rudimentary forms of life are prior in being (*darbūdan*) or creation (*Kashf* V. 2.2). Against this, Sijistānī argues that all three genera exist together, in part for the now-familiar reason that all are equally consequences of heavenly influence (*Kashf* V. 2.3). He also seems to suggest in the course of this argument that plants are created for the sake of animal nutrition. (Despite these arguments, Sijistānī ultimately suspends judgement on the priority question, as the topic belongs to the occult keys (*mafātīḥ-e ġayb*) that are kept with God only, knowledge of which He has kept from (*bī-niyāz kard*) human beings (*Kashf* V. 2.5).) This sheds new light on the previous chapter in the same section (*Kashf* V. 1), where we saw Sijistānī insisting that animal species are never eliminated. For him it is important that plant and animal species are always present, not just because humans benefit from this (though not always: remember the harmful plants and animals) but because it is simply good that the universe contains all possible species. As he says twice in *Kashf* V. 3.1-2, God wanted the animal species to be manifested, so they can neither vanish nor get mixed together, lest "God's wisdom be undermined". From this perspective, there is really no difference between the goodness of animals and the necessity of animals: it is precisely because God wanted them to exist that he set up the universe in such a way that it could not fail to contain them.

## 6. Conclusions

Sijistānī was, as we said at the outset of this paper, an Ismāʿīlī Neoplatonist. We have said a good deal about his Neoplatonism but not much that would be relevant to his Ismāʿīlism. To conclude by rectifying this omission, we would point to a parallel between the place of animals in his philosophy and the place of prophecy in his religious teaching. We have seen that for him, animals are both necessary and beneficial, so that one can understand their emergence from the intelligible realm from two perspectives: by seeing how they are inevitably caused and by seeing why it is good that they be caused. He provides the same sort of naturalistic account for prophecy. There *must* be prophets, because they are a consequence of the emanative scheme, but of course it is also *beneficial* that there should be prophets. Sijistānī draws the parallel himself at *Kashf* VI. 1.3, writing that the functions of elements, celestial, bodies, "the purpose (*qaṣd*) of soul in the plants and the animals", and the workings of soul in humankind all reveal the "dominion and greatness" of intellect (Cf. *Kitāb al-Yanābīʿ*, [15] §164). The flowing of light from intellect onto individual humans who have the perfect "disposition" to receive that light, so as to make prophets, is just another example of the same phenomenon: that the "blessings of

intellect reach soul and nature, giving rise to plentiful things" (compare with *Kitāb Ithbāt al-nubuwwāt* [36] (II. 2, pp. 135–137; V. 9, pp. 263–267)).

**Author Contributions:** All authors have contributed equally. All authors have read and agreed to the published version of the manuscript.

**Funding:** This paper was written under the aegis of the project "Animals in the Philosophy of the Islamic World", funded under the European Union's Horizon 2020 research and innovation programme (grant agreement number 786762).

**Institutional Review Board Statement:** Not applicable.

**Informed Consent Statement:** Not applicable.

**Conflicts of Interest:** The authors declare no conflict of interest.

## Appendix A. Passages on Animals from the *Kashf*

*Appendix A.1. Manuscript Description*

These excerpts have been edited based on four manuscripts and a printed version provided by Henry Corbin. Corbin based his edition on a unique manuscript identified and owned by the scholar and politician Sayyed Naṣr-Allāh Taqavī/Akhavī (d. 1947). Although the Taqavī codex lacks a specific date, ownership notes suggest at least a sale in Rabīʿ al-thānī 804 AH (Nov.–Dec. 1401). The codex itself is much older; based on the archaic form of the naskhī handwriting, the quality of the paper, the state of absorption of the ink, and the orthographic peculiarities, Taqavī and Corbin proposed the end of the sixth century AH (early twelfth century CE) as its likely date (Corbin, "Introduction". In [24] (p. 10)). Corbin faithfully reproduced the orthographical features of the original manuscript in his edition. Despite Taqavī's generous donation of his library and manuscript collection to the Iranian parliament's library (Majles Library), this particular codex was not part of that collection. It appears to have remained in the possession of the Taqavī family, and unfortunately, we have been unable to determine its current location.

In the absence of the Taqavī codex, we used Corbin's text (referred to as "كى" in the apparatus) and the following manuscripts, which were transcribed from the Taqavī codex[1].

MS. Mīnovī 139/1, held in the Ketābkhāneh-ye Mojtabā Mīnovī (d. 1977) in Tehran. Referred to as "مى". Mīnovī transcribed this manuscript himself from the Taqavī codex in May–June 1927, noting the corresponding folio numbers of the Taqavī manuscript. The only difference between the Taqavī manuscript and Mīnovī's transcription is that Mīnovī puts the titles of the Discourses (*maqālat*) and Issues (*jastār*) at the beginning of the lines. This resulted in one extra line in the corresponding pages compared to the Taqavī manuscript (MS. Mīnovī 139/1, page 80, 14–22). It is written in *naskhī* script and comprises 180 pages of 21–22 lines per page. The text of the *Kashf al-Maḥjūb* is found on pages 3–80.

Both Corbin and Mīnovī, in their glosses, have referred to the Taqavī manuscript as "*aṣl*". In this edition, we reproduced such indications correspondingly as "(كى)اصل" and "(مى)اصل". They both replicate the orthographical features of the Taqavī manuscript. Most notably they reproduce the distinction between *dāl* and *dhāl* at the end of words (*dhāl-e muʿjam*) and the omission of silent "h" at the end of a word when its plural form is built with *hā* (e.g., they write بشرها instead of بشرهها).

MS. Adabiyāt 194-jīm, housed in the library of the Faculty of Literature and Human Sciences at the University of Tehran. Referred to as "اد". Transcribed by Moḥammad ʿAlī ʿEbrat Nāʾīnī from the Taqavī manuscript in February 1940, it is written in naskhī script and consists of 113 pages of 17 lines. In April 1940, ʿEbrat Nāʾīnī presented it as a gift to the scholar Sayyed Karīm Amīrī Fīrūzkūhī (d. 1984).

MS. Majles-e Shaurā 11692-aksī, housed in the library of the Iranian Parliament in Tehran. Referred to as "جم". It was transcribed by Ebrāhīm Zanjānī from the Taqavī manuscript in 1931. Our copy has a break in the middle of *Kashf* VI. 6.1 (page 60 according

to the manuscript numbering, which was potentially done by another individual; this corresponds to page 80, line 1 in Corbin's edition). The part before the break consists of 60 pages of 20 lines per page. Subsequently, there is a kind of insertion of 28 pages, offering an incomplete transcription of Nāṣer-e Khosrow's *Goshayesh-o Rahāyesh*. Then, another hand resumes the copy of the *Kashf* (10 pages of 19–20 lines). The pages of this supplement are neither numbered nor in order. The fact that Nāṣer-e Khosrow's *Goshayesh-o Rahāyesh* follows *Kashf* in the Taqavī manuscript suggests that Zanjānī transcribed the entire codex. The first part is in *taʿlīq* script, while the supplement is in *nastaʿlīq*. A duplicate of the deficient MS. Majles-e Shaurā 11692 can be found in Dār al-Kutub, Cairo (MS. Dār al-Kutub, *taṣavvuf fārsī* 81) ([63] (Vol. 1, pp. 829–830)).

MS. Dāneshgāh-e Tehrān 8798, held at the Central Library of the University of Tehran. Referred to as "دت". This manuscript was, like MS. Majles, completed by a second hand. The first part is written in *naskhī* script and comprises 48 pages of 19 lines per page. In this case, the break comes in the middle of *Kashf* V. 6.1 (page 66, line 6 in Corbin's edition), but it is immediately completed by the second hand in 29 pages of 17–18 lines. Both parts are undated, and the identity of the scribes remains unknown. The codex once belonged to Mīnovī.

MS. Malek 4055-2, preserved in the Malek National Library in Tehran. Referred to as "مل". This manuscript was transcribed by Aḥmad Sohaylī Khʷānsārī (d. 1994–5) in 1936. In contrast to preceding copies, it appears to be derived from MS. Majles-e Shaurā 11692, as it is also severed in the middle of *Kashf* V. 6.1. (p. 128, l. 5) and continues with Nāṣer-e Khosrow's *Goshayesh-o Rahāyesh*. However, unlike the MS. Majles, it remains incomplete. MS. Malek 4055-2 is penned in *naskhī* script and consists of 73 pages with 14 lines per page. The text of *Kashf* occupies pages 56–128 of codex 4055, while pages 128–64 include the *Goshayesh-o Rahāyesh*, mistakenly thought to be the continuation of the *Kashf*. Unfortunately, the first treatise in this codex was not accessible to us. This manuscript rarely replicates the orthographical features of MS. Majles and, especially after *Kashf* IV. 6, transcribes the conjunctive and pronoun *kay* (and *-ak*) as *ki^h*/*ke^h2*. In the colophon, Sohaylī Khwānsārī outlines his intention to collate the manuscript, but it seems he did not manage to do so.

In addition to the above, Saʿīd Nafīsī (d. 1966) personally transcribed a copy of the Taqavī manuscript in 1942–3 for his own use; its current location is unknown ([64]).

In this edition, we utilized MS. Mīnovī 139/1 as our primary text, with discrepancies described in the footnotes. Regarding orthography, we did not retain the archaic features present in the original, a departure from Corbin's edition. Specifically, we did not replicate the distinction between *dāl* and *dhāl* at the end of words (*dhāl-e muʿjam*), except in the critical apparatus. The copula is consistently written as a separate word. In the apparatus, text between "[]" indicate the exact phrases in the main body of the text that are subject to variation, while text between "<>" indicate our additions to the text. Our manuscripts rarely indicate vowels (including the *kasra* of *ezāfa*); if they do so and the vocalization affects the meaning (as in *qūt* as opposed to *q^ov^at*), we have indicated them in the apparatus. The vocalization here is added in order to facilitate the reading.

Here is a list of the signs and abbreviations used in the apparatus and their connotation (Table A1):

**Table A1.** The Legend of Symbols and Abbreviations Used in the Edition.

| In the Margin of the Given MS. | Subscribed *ḥā* | MS [word] ح |
|---|---|---|
| On the top of the indicated word ("[word]") in the given MS. | Subscribed *fawq* | فوقMS [word] |
| Corrected by the scribe in the margin | Subscribed *ṣḥ* | صحMS [word] |
| Alternative word for "[word]" offered by the scribe | Subscribed *kh* | خMS [word] |

**Table A1.** *Cont.*

| In the Margin of the Given MS. | Subscribed *ḥā* | MS [word] |
|---|---|---|
| The "[word]" in the given MS. is not legible | Subscribed *nākh^wānā* | ناخوانا MS [word] |
| The "[word]" is absent from the given MS. | M-dash | —: MS [word] |
| In the given MS. the "Phrase" comes after the "[word]" | | Phrase+: MS [word] |
| "⎢⎢" separates multiple occurrences of the same word in a paragraph | | Word ⎢⎢ Word |
| Crossed out by the scribe | | ~~word~~ MS [word] |
| Word$_1$ is crossed out by the scribe and replaced with word$_2$ | | word$_2$ ← ~~word~~$_1$ MS [word] |

*Appendix A.2. Edition and Translation of* Kashf al-maḥjūb

**Index of Contents of**

**Table A2.** The Seven Discourses (*Maqālāt*) of the *Kashf al-maḥjūb*, Each Divided into Seven Issues (*jastār*).

| *Discourse I: On the Unity* | مقالتِ اوّل: در توحید |
|---|---|
| *Discourse II: On Bringing to Mind the First Creation [Intellect]* | مقالتِ دوم: در یاد کردنِ خَلقِ اوّل [= خِرَد / عقل] |
| *Discourse III: On the Second Creation [Soul]* | مقالتِ سوّم: اندر خلقِ ثانی [= نفس] |
| *Discourse IV: On the Third Creation, namely Nature* | مقالتِ چهارم: در خلقِ ثالث و آن طبیعت است |
| *Discourse V: On the Fourth Creation [the types of Existents; Metempsychosis]* | مقالتِ پنجم: در خلقِ رابع [= موجوداتِ روی زمین—تناسخ] |
| *Discourse VI: On the Fifth Creation [Prophethood, Imamate]* | مقالتِ ششم: در خلقِ خامس [= پیغمبری—نبوّت—امامت] |
| *Discourse VII: On Mentioning the Sixth Creation [the resurrection]* | مقالتِ هفتم: در ذکرِ خلقِ ششم [= معاد—قیامت] |

مقالتِ اوّل: در توحید

**Discourse One: On the Unity**

**Table A3.** The Seven Issues of the *First Discourse* dedicated to Sijistānī's Apophatic Theology.

| **I. 1**: *On removing thingness from the Creator* | جستار اوّل: در دور کردنِ چیزی از آفریدگار |
|---|---|
| **I. 2**: *On removing limit from the Creator* | جستارِ دوم: در دور کردنِ حدّ از آفریدگار |
| **I. 3**: *On removing attributes from the Creator* | جستار سوّم: در دور کردنِ صفات از آفریدگار |
| **I. 4**: *On removing place from the Creator* | جستار چهارم: در دور کردنِ مکان از آفریدگار |
| **I. 5**: *On removing time from the Creator* | جستار پنجم: در دور کردنِ زمان از آفریدگار |
| **I. 6**: *On removing being from the Creator* | جستار ششم: در دور کردنِ هستی از آفریدگار |
| **I. 7**: *On removing the opposites of these terms from the Creator [against taʿṭīl and tashbīh]* | جستار هفتم: در دور کردنِ آنچ برابر این لقبها اُفتد از آفریدگار |

<div dir="rtl">

**مقالتِ اوّل در توحید؛ جستارِ اوّل: در دور کردنِ چیزی از آفریدگار**

(۳) دیگر، اگر جایز بودی دربستنِ چیزی در خدای، واجب شدی گفتن کی۳ چیزی آفریدگار است۴ و چیزی آفریده. و این چیزها یا جوهر بَوَد یا عرض، و جوهر یا جسم بَوَد یا روح، و جسم یا۵ نامی بَوَد یا مَوات، و نامی یا نبات بَوَد یا حیوان. [...]

</div>

**Discourse I, Issue 1: On removing thingness from the Creator**

**I. 1.3.** Further, if attributing "thingness" (*čīzī*) to God were permissible, one would have to say that "one thing is the Creator" and "another thing is the creature". And these [later] "things" would be either substance or accident. But substance is either body or spirit, and body is either animate or inanimate (*mavāt*), while animate is either plant or animal.

<div dir="rtl">

مقالتِ دوّم: در یاد کردنِ خلق اوّل

</div>

**Discourse Two: On Bringing to Mind the First Creation**

**Table A4.** The Seven Issues of the *Second Discourse* on the Emanation of the Intellect.

| | |
|---|---|
| **II. 1:** *On the meaning of the [statement that] intellect is the center of both worlds* | جستارِ اوّل: در معنیِ آنک خرد مرکزِ دو جهان است |
| **II. 2:** *That intellect becomes one with the command of God which is expressed as command to oneness* | جستارِ دوم: در یکی شدنِ خرد به اَمرِ ایزد کی عبارت از امر به وحدت کنند |
| **II. 3:** *That intellect becomes one with the Command of God which is expressed as Command to the Word* | جستارِ سوّم: در یکی شدنِ خرد با اَمرِ ایزد کی عبارت از امر به کلمه کنند |
| **II. 4:** *That intellect becomes one with the command of God which is expressed as a command to knowledge* | جستارِ چهارم: در یکی شدنِ خرد با اَمرِ ایزد کی عبارت از امر به علم کنند |
| **II. 5:** *On the meaning of Intellect's becoming one with the command of God which is expressed as command to the command in itself (amr be^h nafs-e amr)* | جستارِ پنجم: در معنیِ یکی شدنِ خرد با اَمرِ ایزد کی عبارت از امر به نفس امر کنند |
| **II. 6:** *How the seed of both worlds is in intellect* | جستارِ ششم: در چگونگیِ آنک تخمِ دو جهان در خرد است |
| **II. 7:** *How the conjoining of intellect with soul [takes place]* | جستارِ هفتم: در چگونگیِ جفت گشتنِ خرد با نفس |

<div dir="rtl">

**مقالتِ دوّم، جستارِ هَفتم: در چگونگیِ جُفت گَشتنِ خِرَد با نفس**

(۱) بدانک نفس با خرد۶ جُفت گردد آنگاه کی۷ چیزها را گِرد کند و آنگاه کی۸ چیزها را بگُشاید، و همچنین آن زمان کی بخششِ چیزها کند. و این چُنان۹ است کی نفس هر گه کی خواهد کی وحدت را بگُشاید تا وحدت بسیار گردد، نتواند گُشادن مگر کی خرد با او جُفت گردد. و هر گه۱۰ کی در گشادنِ چیزها شود، صواب آرَد بدانک به جُفتیِ خرد بود با نفس. و این چنان بَوَد کی جوهرِ مطلق یکی است و در گاهِ گُشادن کی بسیار گردد، جسم و روح شود. پس گُشادنِ نفسُ جوهر را۱۱ کی یکی بود در اوّل، پس۱۲ جسم و روح۱۳ گشت، و درگُشادن، از بهر جفت گشتنِ خرد بَوَد با نفس. پس ایدون گوییم کی۱۴ جسم یکی است در اوّل، و۱۵ از بهر گُشادن به دو قسمت شود: قسمی مَوات و قسمی نبات کی نفس نامی پذیرد. پس گُشادنِ نفس جسم را کی یکی بود در اوّل، پس به دو قسم شد چون موات و نبات، از بهر جفت گشتنِ خرد بود با نفس. پس ایدون گوید کی آنک۱۶ نفس نامی پذیرد به دو قسمت شود: قسمی نبات و قسمی حیوان، و این تمییز۱۷ کردن از بهر۱۸ جفت گشتنِ خرد است با نفس. پس گوید: حیوان به دو قسمت شود: قسمی ناطق و قسمی غیرِ ناطق، و ناطق به دو قسمت شود: یکی میرَنده چون مردم و یکی نامیرَنده چون ملائکه۱۹. و این همه از بهر جفت گشتنِ خرد بود با نفس.

</div>

**Discourse II, Issue 7:** *On the quality of the conjoining of intellect with soul*

**II. 7.1.** You should know that the soul conjoins with the intellect at the moments when it gathers things, and when it unfolds them, and likewise when it divides things. And this occurs in such a way that, each time that soul wants to unfold the unity so that the unity becomes many, it can unfold [the unity] only if the intellect conjoins with it. And whenever it begins to unfold things, it brings about what is right (*ṣavāb*) due to the conjoining of the intellect with the soul. This [unfolding] is such that the absolute substance (*jawhar-e moṭlaq*)

is one; but at the moment of unfolding, when it becomes many, it becomes body and spirit (*rūḥ*). Thus, the soul's unfolding of substance—which is one at first but then becomes body and spirit—and unfolding [in general] (*dargoshādan*), is because of the conjoining of the intellect with the soul.

So we say that body is at first one, then because of the unfolding it divides into two: one is the inanimate (*mavāt*) and the other is the animate (*nabāt*)[20], which receives the growing soul (*nafs-e nāmī*). Thus, the soul's unfolding of body—which was one at first but then then divided into two as inanimate and plant—is because of the conjoining of the intellect with the soul. Then, in a similar manner, it is said that what receives the growing soul divides into two: one division is plant, the other is animal. This distinction occurs because of the conjoining of the intellect with the soul. Further, it is said that the animal divides into two: one division is rational (*nāṭiq*) and the other is non-rational (*ġayr-e nāṭiq*). The rational, [in turn,] divides into two: one mortal (*mīrandeʰ*), like human being (*mardom*), the other immortal, like the angels. And all this is because of the conjoining of the intellect with the soul.

(۲) دیگر، هر گه که نفس خواهد کی[21] چیزهای پراکنده را گِرد کند و در یکی چیز بندد، نتواند مگر به[22] جُفت گشتنِ خرد با وی. و مثالِ این چنان است کی گوید: ناطق دو[23] بهر است: میرنده و[24] نامیرنده؛ و میرنده دو بهر است: بهری ناطق و بهری غیر ناطق[25]؛ و غیر ناطق[26] دو بهر است: بهری حیوان و بهری نبات؛ و نبات دو بهر است: بهری نامی و بهری موات؛ و موات دو بهر است: جسم است و جز از جسم؛ و جسم به دو بهر است: جوهر و جز از جوهر، چون جوهر اوّل. و بدانک این پَراکنده‌ها[27] کی نفس گِرد آورَد و از بسیاری با یکی جوهر بَرَد چنانک از یکی جوهر بسیاری آورَد، به جُفت گشتنِ خرد بود با نفس. *فاعرفه*.

II. 7.2. Further, whenever the soul wants to bring together things that are scattered and join them into a single thing, it can do so only by conjoining with the intellect. Take the following example: it is said that the rational is divided into two, mortal and immortal. The mortal is again divided into two, rational and non-rational. The non-rational is divided into animal and plant, and plant into growing and inanimate. The inanimate is divided into body and non-body. The body divides into substance and non-substance, such as the first substance. You should know that the soul's bringing together scattered things into one substance from many, just as it brings many from one substance, occurs because of the conjoining of the intellect with the soul. Understand this!

مقالتِ سوّم: اندر خلقِ ثانی

## Discourse Three: On the Third Creation

**Table A5.** The Seven Issues of the *Third Discourse* on the Emanation of the Universal Soul.

| | |
|---|---|
| **III. 1:** *That the Soul has descended in the form of man* | مقالتِ سوّم: اندر خلقِ ثانی |
| **III. 2:** *That the motion of soul collects/gathers all motions* | جستارِ اوّل: در آنک نفس به صورتِ مردم فروآمده |
| **III. 3:** *Knowing the division of plant within the Soul* | جستار دوم: در آنک حرکتِ نفس گردآور همهٔ حرکات است |
| **III. 4:** *Knowing the share of the animal in Soul* | جستار سوّم: در شناختنِ بخشِ نبات از نفس |
| **III. 5:** *How sense perception depends on the sensory perceiver which is the soul* | جستار چهارم: در شناختنِ بهر حیوان از نفس |
| **III. 6:** *How discourse (noṭq) depends on the speaker, which is the soul* | جستار پنجم: در چگونگي بازگشتنِ حسّ به حاسّ و آن نفس است |
| **III. 7:** *How thought depends on the thinker, which is the soul* | جستار ششم: در چگونگي بازگشتنِ نطق به ناطق و آن نفس است |

مقالتِ سوّم، جستارِ اوّل: در آنک نفس به صورتِ مردم فُروآمده

(۱) آگاه باش کی نفس به صورتِ مردم فروآمد از جملهٔ چیزهای عالم، زیراک[28] نفس در زندگانی پیوسته شود و در اعتدالِ قرار گیرد، و اندر عالمِ طبیعی هر چه دونِ مردم است[29] اعتدالِ تمام نپذیرد. و چیزهای[30] عالم به سه بهر است[31]: یکی موات و یکی نبات و یکی حیوان. پس نفس را فُروآمدن و پیوستن نیست بدان دو قسمت، زیراک[32] نفس در زندگانیِ حسّی و در اعتدالِ عقلی پیوندد، و این هر دو بهر[33] در مردم است، زیراک[34] زندگانی[35] لُبِّ طبیعت است، یعنی مغز، و اعتدالِ لُبِّ عقل است، از بهر آنک[36] نفس را یک کرانه با عقل است و یک کرانه با طبیعت، و او را قرار نبوَد مگر اندر[37] میانِ هر دو، و آن زندگانیِ طبیعت است[38] و اعتدالِ عقل. ازین[39] جهت[40] گفتیم کی نفس به صورتِ مردم فروآمد. *فاعرفه*.

### Discourse III, *Issue One: That the Soul has descended in the form of human*

III. 1.1. You should be aware that, among all things of the world, the soul has descended in human form because the soul joins life and dwells in harmony; and in the natural world, nothing inferior to the human receives complete harmony. And things in the world are divided into three: inanimate, plant, and animal. So there is for the soul no descending to and joining to those [other] two divisions, because the soul joins the sensible life and the intellectual harmony, and both are in the human. For life is the kernel (*lobb*) of nature, that is, its nucleus (*maghz*)[41], while harmony is the kernel of intellect. This is because the soul has one border with the intellect and another border with nature, and dwells only between these two, and it consists in the life of nature and the harmony of intellect. This is why we said that the soul has descended in the form of man. Understand this!

مقالتِ سوّم، جستار سوّم: در شناختنِ بخش نبات از نفس

(۱) چون کارِ عالمِ جسمانی و روحانی بر دوگانگی قرار یافت[42]، و آن فایده دادن است و فایده پذیرفتن، واجب آمد کی هرچه از موالید است اندرو یابی فایده دادن و فایده پذیرفتنِ[43] طبیعی. و سختتر قبولی بوَد نبات را از جملهٔ موالید[44]، و سختتر فایده‌ای[45] از عناصر. و نبات[46] سختترین فایده‌ای[47] کی دهد، حیوان را دهد، زیراک بقای حیوان در آن چیزها بود کی از نبات ظاهر شود از گیاهها و میوه‌ها[48]. و چون یافتیم در نبات فایده دادن[49] و فایده پذیرفتنِ[50] طبیعی، حکم کردیم آنگاه کی نبات را بهر است از نفس شریف. بدین برهان هرچند واجب آمد بهر نبات از نفس، واجب آمد کی فایده دادنِ نبات، آنک لطافتِ نبات است و مغز او است، مردم را بود، و آنک کثافت بود حیوان را بود، هر یکی را[51] بر مقدار لطافت و کثافتِ ایشان. و بدانک آنک دونِ نبات است، چون معادن، فایده پذیرفتنِ[52] اندرو ظاهر است و فایده دادن[53] اندرو نیست، و نسبت نکردند بهر نفس را با او، مگر با نبات از بهر فایده دادن و فایده پذیرفتن، و ازین قِبَل[54] نفس نامی را اضافت با نبات کردند کی نبات هَمی زیادتی گیرد از فایده پذیرفتن، و از فایده دادن زیادتی پدید آورد در قابلانِ خود. *فاعرفه*.

### Discourse III, Issue One: *Knowing the division of plant within the Soul*

III. 3.1. As the work of both the corporeal and the spiritual universe is twofold—it consists in both bestowing and receiving benefits—the natural bestowal and reception of benefits became necessary in all generated things. Among these, the greatest reception is found in plants, the greatest [bestowal of] benefit for the elements. The maximum benefit that plants give is to the animals, because animal survival depends on the vegetables and fruits taken from plants. Having found both natural bestowal and reception of benefit in plants, we judged that plants participate in the Noble Soul (*nafs-e sharīf*). As a corollary of this demonstration, as the participation of plants in soul became necessary, it was [also] necessary that plants bestow benefit to the human with its more subtle aspect and nucleus, while its denser aspect is for the animal: for each in proportion to its degree of subtlety and density. You should know that in what is inferior to the plant, like minerals, one can also observe the reception of benefits, but they do not bestow benefits. So one can draw no relation (*nesbat*) in their case in terms of participation in soul, unless [one compares them to] plants with respect to giving and receiving benefits. The reason why the growing soul (*nafs-e nāmī*) is related to plants is that the plants grow constantly due to the benefit they receive; in turn, by bestowing benefit, they cause growth in those who receive them. Understand this!

(۲) و دیگر نبات را[55] از نفس بهری شریف است و آن مشاکلت است کی میانِ مغز نباتها است[56] و میانِ زندگانیِ[57] طبیعی[58]، کی کسب[59] کند زندگانیِ طبیعی را تا آثارِ نفس ظاهر شود و غریزتِ عقل پیدا گردد. و اگر در نبات این بهر شریف نبودی و مشاکلتش نبودی با زندگانیِ طبیعی، طبیعت را ازو غذا گرفتن نشایستی. بدین دلیل روشن شد بهر نبات از نفس. *فاعرفه*[60].

III. 3.2. Further, there is yet another noble participation in soul for plants, namely the resemblance (*moshākelat*) that exists between the nucleus of plants and natural life (*zendegānī-ye ṭabīʿī*). Plants acquire natural life so as to display the effects of the soul and manifest the innate character (*gharīzat*) of the intellect. If plants lacked this noble participation in soul, and had no resemblance to natural life, nature would not fittingly draw nourishment from [plants]. This indication reveals the participation of plants in soul. Understand this!

مقالتِ سّوم، جستار چهارم: در شناختنِ بهرِ حیوان از نفس

(۱) آگاه باش کی در۶۱ میانِ عقل و نفس دو قوّتِ دیگر است و آن محبّت و غلبه است. به محبّت فایده دهد عقلْ نفس را، و به غلبه فایده پذیرد نفس از عقل. و این دو قوّت از نفس فُروریخت بر حیوان بیرون از آن۶۲ دو قوّت کی فروریخته بود بر ایشان، و آن دادن است و پذیرفتن. نبینی۶۳ هیچ حیوانی را الّا کی دوست دارد جفت و فرزندِ خویش را به محبّتِ طبیعی، و دوست دارند کی غلبه کنند بر یکدیگر۶۴. و ظاهر است فایده گرفتنِ حیوان از نبات؛ و فایده دادن حیوان بدان چیزها پدید آید از منفعت‌ها چون شیر و گوشت و روغن و پشم و موی. و این حالی ظاهر است از بهر آنک غایتِ قوّتِ حیوان این پنج حسّ است و دریافتنِ محبّت و غلبه درین پنج حسّ. محبّتِ بصر چون دوست داشتن لونهای نیکو و رویهای نیکو بود بصر را، و غلبتِ بصر۶۵ دریافتِ صورت‌هایی بود کی نفس۶۶ خواهد کردن. محبّتِ سمع چون دوست داشتنِ آوازهای خوش و حکایاتِ خوش، و غلبتِ سمع بیرون آوردنِ لحن‌ها از موسیقار. محبّتِ شمّ چون دوست داشتن بویهای خوش، و غلبتِ شمّ۶۷ چون کشیدن نَسیمِ هوا را به خویشتن. محبّتِ ذوق چون دوست داشتنِ طعمهای خوش۶۸ شیرین را، و غلبتِ ذوق چون بگُواران‌یدن۶۹ طعام. محبّتِ لمس چون دوست داشتنْ چیز‌های نرم را و لباس‌های لطیف را، و غلبتِ لمس صحبت کردن و دادنْ نطفه را. این است بهرِ حیوان از نفس. فاعرفه.

**Discourse III, Issue Four:** *On Knowing the share of the animal(s) in the soul*

III. 4.1. You should realize that between the intellect and the soul, there are two other powers, namely love (*maḥabbat*) and dominion (*ġalaba*). Through love, intellect gives benefits to the soul; through dominion, the soul receives benefits from the intellect. These two powers poured down from the soul to the animal in addition to those two [other] powers that had [already] poured down to them, namely the bestowal and receiving [of benefit]. You will see no animal that fails to show natural love for its mate and child; and [animals also] love to dominate each other. The animal's reception of benefits from the plant is obvious, while the animal's bestowal of benefits appears through such beneficial things as milk, meat, butter, wool, and fur. This is a manifest, because the ultimate level (*ġāyat*) of animal power is these five senses, and the grasp (*daryāftan*) of love and dominion is in [all] five. The love [corresponding to the sense of] sight is, for example, being fond of nice colors and lovely appearances (*rūy-hā*), while the dominion of sight amounts to perceiving the forms (*ṣūrat-hā*) that the soul wants to illustrate them. The love of hearing is, for example, being fond of pleasant sounds and pleasant tales, while the dominion of hearing consists in the production of melodies from the pan flute (*mūsīqār*). The love of smelling is, for example, being fond of pleasant fragrances, while dominion of smelling is, for example, inhaling the breeze. The love of taste is like being fond of sweet, delightful savors, while the dominion of taste is like preparing and seasoning the food. The love of touch is, for example, being fond of soft things and fine clothes, and the dominion of touch is like taking [love] companions (*ṣohbat kardan*) and emitting semen. This is the share of the animal in the soul. Understand this!

(۲) و حیوان را از نفس بهری دیگر است و آن آن است کی بهری از حیوان هَمی آموزند ادبِ مردم چون ریاضتِ سُتورانْ حرب کردن و بازی کردن را، و آموختنِ گاو کِشت کردن را، و آموختنِ پیل حَرب۷۰ را، و همچنین در باز و شاهین و یوز شکار کردن را، و طوطی سُخن گفتن را. و اگر ایدونک در حیوان از نفس بهری نبودی کی آن۷۱ محبّت و غلبه است، در ایشان این ادبها نیافتندی.

III. 4.2. There is for the animal another way of participating in the soul, namely that some animals can be educated by humans (*adab-e mardom*), for example, training beasts of burden for fighting and games, training cows for farming, and elephants for fighting. Likewise, in the case of the hawk, the falcon, and the cheetah, which learn hunting, while parrots learn to talk. If animals had did not participate in soul, which consists in love and dominion, one would not find these kinds of human education in them.

(۳) و از نفس۷۲ در حیوان بهری دیگر است شریف‌تر ازین بهرها، و آن رغبتِ ایشان است در بقاء و
گریختن از بیم۷۳ فناء، زیراک۷۴ هیچ حیوان نیست کی به خاصّیت از مرگ نمی‌ترسد و حیوة۷۵ را نمی‌کوشد، و
از بهر این بود کی تباه نمی‌کردند حیوان را بی منفعتی کی بود. *فاعرفه*.

III. 4.3.  There is still another way of partcipating in the soul in the animal, nobler
than these, which is their longing for survival (*baqā⁾*) and fleeing out of fear of destruction
(*fanā⁾*), because there is no animal that does not by its own nature fear death and seek
survival. It is for this reason that one does not kill an animal unless there is a benefit [in it].
Understand this!

مقالتِ چهارم: در خلق ثالث و آن طبیعت است

**Discourse IV: On the Third Creation, which is Nature**

**Table A6.** The Seven Issues of the *Fourth Discourse* on the Emanation of the Nature.

| | |
|---|---|
| **IV. 1:** *That the beings can be perceived through Nature* | جستارِ اوّل: در آنک هستی‌ها به طبیعت در توان یافت |
| **IV. 2:** *That Nature does not change its state* | جستارِ دوم: در آنک طبیعت از حالِ خویش بنَگردَد |
| **IV. 3**: *That natural forms are not posterior to the natural principle* | جستارِ سوّم: در آنک صورت‌های طبیعی نه از پس‌تر است از اصل‌های طبیعی |
| **IV. 4:** *That the center of nature is closer to neighboring upon the spiritual realm* | جستارِ چهارم: در آنک مرکز طبیعت سزاوارتر است به همسایگی کردن با روحانیان از آفاق |
| **IV. 5:** *How nature receives assistance from the Soul* | جستارِ پنجم: در چگونگی مدد کشیدن طبیعت از نفس |
| **IV. 6:** *That the beauty of nature, namely the adornment of nature, is spiritual* | جستارِ ششم: در آنک زینتِ طبیعت یعنی آرایش طبیعت روحانی است |
| **IV. 7:** *That composite, generated things emerge from the "mothers" (i.e., elements) and "fathers" (i.e., celestial spheres) through God's providence* | جستارِ هفتم: در آنک موالید از اُمّهات و آباء ظاهر نشود مگر به تدبیر ایزد |

**مقالتِ چهارم؛ جستارِ ششم: در آنک زینتِ طبیعت، یعنی آرایش طبیعت، روحانی است**

(۳) و دیگر کی چون نفسی ماهر گردد در صنعتِ۷۶ نگارگری، تواند حکایت کردن از آرایش آن۷۷ و
رنگ‌های آن۷۸، تا اگر خواهد، پدید آورد صورتِ آن چیزها کی خواهد، چون صورتِ حیوانات یا صورتِ تخت
و خانه یا صورتِ مردم. و این حکایت کی او کند از بهر آن است کی آن روحانی است و مشاکلتِ۷۹ او به جوهر
روحانی۸۰ نبینی،۸۱ کی هیچ کس نتواند چیزی طبیعی کردن نه حیوانی و نه نباتی و نه معدنی، ولیکن۸۲ تواند
پیدا کردن آنچ۸۳ خواهد از آرایش طبیعت، و از بهر آن تواند کردن کی آن روحانی بود. *فاعرفه*.

**Discourse IV, Issue Six:** *That the beauty of nature, namely the adornment of nature,
is spiritual*

IV. 6.3.  Further, when a soul becomes skilled in the art of painting, it can imitate
(*ḥekāyat kardan*) the beauty of that [natural thing] and its colors, so that if it wishes, it can
bring into appearance the form of those things that it intends. For instance [it can represent]
the form of animals, or the form of a bed and a house, or the form of human. This imitation
it fashions is due to the fact that [the form] is spiritual (*rūḥānī*), but you do not see its
resemblance (*moshākelat*) to the spiritual substance. [And it imitates] because nobody can
create a natural thing, neither animal, nor plant nor mineral; however, one can render
visible (*paydā kardan*) that which one intends of the adornment of nature, and one can do so
because that [adornment] is spiritual. Understand this!

**مقالتِ پنجم: در خلقِ رابع [=موجوداتِ روی زمین—تناسخ]**

**Discourse V: On the Fourth Creation**

**Table A7.** The Seven Issues of the Fifth Discourse dedicated to the Sublunary, Terrestrial Generated Things and Metempsychosis.

| | |
|---|---|
| **V. 1:** *That the species are preserved* | جستارِ اوّل: در آنک انواع نگاه داشته است |
| **V. 2:** *That Animals did not appear after Plants* | جستارِ دوم: در آنک حیوان نه از پس نبات پدید آمد |
| **V. 3:** *That the species do not mix with each other, neither at composition, nor after composition* | جستارِ سوّم: در آنک انواع با یکدیگر درنیامیزند، نه در ترکیب و نه پس از ترکیب |
| **V. 4:** *That not one of the Species is destroyed, and none is added* | جستارِ چهارم: در آنک انواع یکی باطل نگردد و یکی افزون نشود |
| **V. 5:** *That the increase and decrease of existing individuals does not entail either growth or decrease in virtues* | جستارِ پنجم: در آنک بسیاری و کمی افراد در فضایل افزونی و کمی نیارد |
| **V. 6:** *That the circular motion, namely spherical motion, is the cause of animal motion* | جستارِ ششُم: در آنک حرکتِ مُدوّر یعنی جُنبِش گِرد علّتِ حرکاتِ حیوانات است |
| **V. 7:** *That the number of species is limited* | جستارِ هفتم: در آنک انواع معدود است |

<div dir="rtl">

مقالتِ پنجم؛ جستارِ اوّل: در آنک<sup>84</sup> انواع نگاه داشته<sup>85</sup> است

(۱) اگر جایز بودی کی<sup>86</sup> یک نوع از نگاهداشتن<sup>87</sup> بیرون شدی و بازپس أُفتادی، جایز بودی کی همه انواع از نگاه داشتن<sup>88</sup> بیرون شدندی و همه بازپس افتادندی. و اگر چنین بودی، فساد همه را بهباطل کردی. و اگر فساد همه را بهباطل کردی، بهباطل بودی آنک در زیرِ انواع است و آن اشخاص است، و بهباطل بودی آنک زَبَرِ<sup>89</sup> انواع است و آن اجناس است، زیراک<sup>90</sup> اجناس گِرد آورد انواع را و انواع گِرد آورد اشخاص را. و بهباطل گشتنِ<sup>91</sup> اشخاص باطل شدنِ عالمِ سفلی بوَد، و بهباطل گشتنِ اجناس باطلِ شدنِ عالمِ علوی باشد؛ و این هیچ دو<sup>92</sup> در عالم باطل نیست. پس درست شد کی اجناس و اشخاص باطل نگردد. پس چون واجب آمد بماندنِ اجناس و اشخاص، واجب آمد بماندنِ انواع کی<sup>93</sup> هر دو را نگاه دارد<sup>94</sup>. درست شد کی انواع نگاه داشت<sup>95</sup> است و یکی باطل نگردد. فاعرفه.

</div>

**Discourse V, Issue One:** *That the species are preserved*

V. 1.1. If any one species could be excluded from preservation (*negāh-dāsht*) and so vanish (*bāz-pas oftādī*), then all species could be excluded from preservation, and so would all vanish. In that case corruption would destroy all of them. And if this happened, then what falls under the species—namely the individuals—would also be destroyed, so that which is above the species—namely the genera—would be destroyed too, because the genera gather the species [under themselves] and the species gather the individuals [under themselves]. And the destruction (*be^h-bāṭel gashtan*) of individuals is equivalent to the destruction of the lower world, while the destruction of the genera is equivalent to the destruction of the higher world. In fact though, neither of them is destroyed, so neither individuals nor genera are destroyed. Thus, since the persistence (*bemāndan*) of both the genera and the individuals has been shown to be necessary, the persistence of the species, which preserve both, is necessary too. It has then been established that the species are preserved, and not even one of them is destroyed. Understand this!

<div dir="rtl">

(۲) و دیگر کی انواغ شعاعاتِ آن صورتها است، کی مرکز آن افلاک است، کی نهایتِ برشدنِ طبیعت است. و نیست چیزی ازین چیزها کی باطل گردد، بلک آن نگاه داشته است در اجرامِ علوی. و نیست میانِ آن

و میانِ مرکز مانعی کی انواع بر آن قرار گرفته دارد. پس ازین جهت واجب است<sup>96</sup> پایندگی صورتهای فلک، و ازین جهت واجب کند کی میان کواکب و میانِ انواع مانعی نیست. ازین جهت<sup>97</sup> واجب شد کی<sup>98</sup> انواع نگاه داشته<sup>99</sup> بود.

</div>

V. 1.2. Further, the species are the radiations of those forms, whose center is the celestial spheres that are the outermost, ascending limit of nature. There is nothing among these things that would be destroyed; rather they are preserved in the celestial bodies, and there is between them and the center [of the universe] no obstacle at which the species would come to settle. So in this way, it is shown to be necessary that the forms of the sphere

persist, and that there is no obstacle between the stars and the species. Thus in this way, it is shown necessary that the species are preserved.

(۳) و اگر کسی ظنّ ایدون برد کی انواع افزون‌تر است از صورت‌های فلکی، این صواب نیست. و این ظنّی فاسد است درین حکم، زیراک او چنین داند کی چیز‌های بسیار نوع‌های بسیار است، و آن یک نوع است؛100 مثلِ کبریت و زاج و زر و سیم وعلیٰ هذا یک نوع است؛ و این اختلافِ این چیز‌ها از اختلافِ مزاجِ این جای‌ها است کی این چیز‌ها به تولّد از آن پدید آید. و دیگر، نباتات همه یک نوع است با اختلافِ بسیار کی اندر ایشان موجود است. و همچنین کثرتِ حیوان چون مور و پشه و اسپ و اشتر، کی همه به صورت مختلف اند، یک نوع است. و اگر ایدونک101 ایزد بگشاید بر خلقْ صورت‌های فلک، تواند بگماشتن صورت‌های او کی به تولید همی زاید یک یک. *فاعرفه*.

V. 1.3. If somebody believes that the species are greater in number than the celestial forms, this is not correct. This is a false belief, in which there is a false assumption (*ḥokm*), because the person would be supposing that the many things belong to many species, when in fact there is just one species. For instance [different things like] sulphur, vitriol, gold, silver and so on are one species. The differences between these things is due to the difference in the mixture (*mezāj*) of those places where they arise through generation (*tavallod*). The plants too are all of one species, despite the great many differences that exist among them. The same for the many animals, like ants, gnats, horses, and camels: while different in form, they are all one species. If God thus opens the way (*begoshāyad*) for the forms residing in the sphere to [administrate] the creation, he can also appoint [the sphere's] forms [to oversee reproduction], so that the forms constantly give birth (*hamī zāyad*) [to the individuals] one after another, by way of generation (*tawlīd*). Understand this!

(۴) و دیگر کی انواع بر دو گونه است: یک گونه بر فصلِ اوّل است کی درو مصلحتِ عالم بسته شد؛ و دیگر دربسته شود در آن چیز کی بر102 فصلِ دوم است، کی نه جایز بود کی خلافِ آن بود. امّا آنچ103 بر فصلِ اوّل افتاد از آفرینش حیوان فصلِ مردم است و اسپ و گاو و گوسفند و مرغ. و امّا از نبات، چون گندم و جو و برنج و نیشکر و آنچ104 بدین ماند. امّا105 آنچ106 دربسته شد به حیوان کی بر فصلِ دوم است، چون سباع. و امّا آنچ107 در نبات بسته شد، چون نباتی کی گنده باشد و به بوی ناخوش بود. و امّا آنک108 در معادن بسته شد، چون آب‌های کبریتی و زاج‌ها بود. و ازین چیزی نیست کی نه109 نگاه داشته است، و نگاه داشتنِ او کمتر نیست اندر طبیعت از آن چیز‌ها کی جفتِ ایشان اند و بر فصلِ اوّل اند، زیراک این چیز‌ها اثر هم از آن اصل110 پذیرند کی آن چیز‌ها پذیرند کی در آفرینش بر فصلِ اوّل اند، از بهر آنک این اثر از آن شعاعاتِ صورت‌های فلکی پذیرند111 کی نگاه دارندۀ112 انواع است. واجب شد کی113 انواع نگاه داشته است. *فاعرفه*.

V. 1.4. Further, species are of two kinds: one kind belongs to the primary division (*faṣl*), on which the benefit (*maṣlaḥat*) of the world depends (*darū . . . baste^h shod*). The other is connected (*darbaste^h*) to the things that belong to the second division, which cannot be otherwise than they are. In the creation of animals, what belongs to the first division is the dividing of man, horse, cow, sheep, bird, [etc.]. Among plants they are wheat, barley, rice, sugar cane, and whatever resembles these. Among minerals they are gold, silver, ruby, emerald, iron, lead, and whatever resembles these. As for what is connected to animals and belongs to the second division, this is [for example] beasts of prey, and among plants, those that are fetid and stink. Among minerals they are, for example, sulfurous waters and vitriols. There is nothing among them that is not preserved, and their preservation is no less in nature than that of their counterparts that are connected to the first division. For these things too receive effects from the same principle as bestows effects on those things that in their creation are connected to the first division. For they receive these effects from those [aforementioned] radiations of celestial forms, which are the preservers of the species. It follows necessarily then that [all] species are preserved. Understand this!

مقالت پنجم: جستار دوم: در آنک حیوان نه از پس نبات پدید آمد

(۱) اگر روا بودی کی موالید حرا>114 بهری پیش بودی، حیوان سزاوارتر بودی در پیش115 بودن. آگاه باش کی گروهی از مردمان حکم کردند کی حیوان از پس نبات پدید آمد، و چنین گفتند کی مردم پس از حیوان پدید آمد. و ایشان را چنین اندیشه افتاد کی چون حیوان بی غذا و قوت نباشد، نبات باشد، بعد از آن حیوان. خواستیم116 کی117 بدانیم درین118 جَستار راست و دروغِ این سخن و پیدا کنیم اندرین باب آنچ119 حقّ است.

**Discourse V, Issue two:** *That animals did not appear after plants*

V. 2.1. If there could be a sub-division (*bahre*) for the generated thing (*mavālīd*) that is prior [to the rest], then it would be more appropriate (*sazāvārtar*) for animal to be prior. You should be aware, however, that one group of people judged that animals have appeared after plants; and they argued that humans appeared after animals. The reasoning that occurred to them was that, since animals cannot exist without food and nutrition (*qūt*), plants had to exist first, and only then the animals. In this chapter, we intend to discover whether this doctrine (*sokhan*) is right or wrong, and to find out the truth regarding this topic.

(۲) ایدون گوییم کی یافتنِ همه چیزها[120]، از نبات و حیوان و مردم، ظاهراً موجود است بی شکّ، و پیش[121] بودنِ یکی بر یکی نه موجود است. پس برهان به کار باید[122] آن کس را کی این دعوی کند. همچنانک[123] قوتها مقدّم است بر حیوان، و دارندهٔ حیوان است، از حیوان خاصّه مردم مدبّرِ[124] نبات است، خاصّه آن چیزها را کزان[125] غذا توان گرفتن. و این[126] کسان کی این دعوی کنند، باید[127] کی بهظاهر کنند دربودنِ نبات پیش[128] از حیوان، و دیگر کی پیدا کنند کی[129] از نباتها کدام پیشین بود، زیراک حیوان مختلف بسیار است. اگر ایدونک حکم کنند بر برخی از[130] نبات کی پیشتر بود، پس یا باقی نبات پیشتر از باقی حیوان بود، یا باقی حیوان پیشتر از باقی نبات بود. اگر ایدون گویند کی باقی نبات پیشتر از باقی حیوان بود، دربودنِ[131] آن[132] نباتِ پیشین کفایت بود حیوان را، پس پدید آمدن[133] نبات نه از بهر حیوان است. و اگر ایدونک[134] باقی حیوان پیش از باقی نبات بود، پس حکم بکردند به بودن[135] حیوان پیش[136] از نبات. و چرا این حکم کی بر بهری بکردند[137]، بر کلّ نکردند؟ پس محال است و ناممکن، و این را در برهان جایگه نیست. *فاعرفه.*

V. 2.2. We say then that all things' being found to exist (*yāftan*), including plants, animals and humans, is evidently a fact (*mawjūd*) without any doubt, whereas the priority of one over another is not. Therefore, he who wants to make this claim must offer a demonstration (*borhān*) for it. Just as nutrition (*qūt*) is prior to animals and is what maintains (*dārande^h*) animal [life], so among animals, it is specifically the human who is the caretaker (*modabber*) of plants, especially those plants from which one can get food. So those who make this claim must show that plants came before animals; moreover, they must explain which of the plants is prior, because animals are very diverse. In light of this, if they would judge some of the plants to be prior to some of the animals, then, then the other plants might [also] precede the other animals, or alternatively the other animals might precede the other plants. If they say that the other plants are prior to the other animals, then the existence (*darbūdan*) of the former portion of plants already suffices for [the maintenance of] the animals; so [it would follow that] the appearance of plants is not for the sake of (*na^h az bahr-e*) animals. If on the other hand the other animals were prior to other plants, then the opponents would have admitted that animals are prior to plants in existence. So why not say of the whole what is being said of only some? Hence, it follows that this [initial claim] is absurd and impossible, and there is no place for it in a demonstration. Understand this!

(۳) و دیگر کی قوتِ حیوانی سختتر است از قوتِ نباتی و خلاف نیست کی قوتها به هر دو کشیده است از فلک[138]. به چه علّت واجب آمد کی ضعیفتر قوتِ فلکی بیشتر به خویشتن کشد و پیشتر[139] بود، و قویتر کمتر کشد و پستر بود؟ و از چه چیز بازداشته شد[140] قوتِ قوی را کشیدن مادّهٔ خویش؟ و چه فراپیش[141] برد قوت[142] ضعیف را، و فلک بر همه حرکت همی کند؟ پس درست شد کی بودنِ حیوان پس از نبات باطل است. *فاعرفه.*

V. 2.3. Further, animal nutrition (*qūt*) is more strengthening than the nutrition (*qūt*) of the plant, but there is no difference [between them] insofar as the nutrition of both (*qūt-hā*) is drawn (*keshīdeh*) to them from the celestial sphere. Now, why should the weaker attract the celestial nourishment (*qūt*) to a greater degree (*bīshtar*) to itself, and be prior [in existence] (*pīshtar*), while the stronger would attract lesser [celestial nourishment] and be posterior? And why would the nutrition (*qūt*) of the stronger be prevented from drawing its material to itself? And what has pushed forward the nutrition (*qūt*) of the weaker, even though the sphere constantly moves over them all in the same way? Thus, it has been established that it is false to say that animals come after plants. Understand this!

(۴) و دیگر، اگر روا بود کی گویی «بعضی از موالید پیش‌تر143 بود از بعضی حیوان»، بدان سزاوارتر
بود خاصّه مردم، زیراک144 مردم را همی بینیم کی حیلتِ صلاح بدنِ خویشتن همی کند145 و گِرد همی کند146
از بهر خویشتن و از بهر حیوان چیزها کی منفعت باز147 دهد از غذاها و دیگر چیزها. و حاجتِ حیوان به
صناعت، یعنی پیشه‌ها، کمتر از آن است کی حاجتِ مردم به غذاها، و این آن چیز است148 کی پیشه‌ها149 را
قدیم کند بدین قول.

V. 2.4. Further, if one could say, "Some generated things are prior to some animals",
then it would be more appropriate (*sazāvārtar*) for the human [to be prior]. For we see that
humans (*mardom*) constantly devise contrivances (*ḥīlat … hamī konad*) for the utility and
upkeep (*ṣalāḥ*) of their own bodies, collecting both for his own sake and for the sake of
animals things that will produce benefit, such as food, among other things. [Furthermore]
the animals' need for the crafts, meaning arts (*pīshe^h-hā*), is less than the human need for
food; so by the opponents' argument the crafts should be eternal (*qadīm*), [which is absurd].

(۵) و آگاه باش کی این علمی است150 کی ایزد اهلِ خرد و علم را بی نیاز کرد از تکلیفِ شناختنِ این،
زیراک این ‹حاز›151 آن مفاتیح غیب است، کی ایزد داند و کس نداند؛ و ایشان را بر آنک152 بر ایشان است153،
شناختن سزاوارتر است از شغل بردن154 و شناختنِ آنچ155 بر ایشان ننهادندند156. درست شد کی حیوان نه از پس
نبات بود.157 *فاعرفه*.

V.2V. 2.5. You should be aware that this is a knowledge which God has discharged
the wise and knowing from the obligation of knowing it. For it belongs to the keys of the
hidden (*mafātīḥ-e ġayb*), which God alone knows. It is more appropriate that [the wise]
devote themselves to things that are [essential] for them to know, than for them to strive
after knowledge that is not required. It has then has been established that animals are not
posterior to plants. Understand this!

**مقالتِ پنجم، جستارِ سوّم: در آنک انواع با یک‌دیگر درنیامیزند158 نه در ترکیب و نه پس از ترکیب**

(۱) زیراک159 انواع در اشخاص بسته شد، و اگر جایز بودی کی انواع درآمیختی با یک‌دیگر، جایز
بودی کی اشخاص درآمیختی با یک‌دیگر161 و یافتندی مردمی162 را کی نیمی خَر بودی و نیمی مردم، و خر
نیمی مرغ بودی و نیمی خر، و این نه موجود است و ممتنع است. و از بهر آن نه موجود است کی منافع آن
همه باطل می‌شود و ایزد هیچ چیز را باطل نکند. و از بهر آن ممتنع است درآمیختنِ انواع با یک‌دیگر کی انواع
در اشخاص بسته شدند163. درست شد کی انواع با یک‌دیگر164 درنیامیزند165. *فاعرفه*.

**Discourse V, Issue Three:** *That the species do not mix with each other, neither in*
*composition, nor after composition*

V. 3.1. This is because the species are connected to the individuals. Now, if the species
could mix with each other, then individuals [of different species] could mix with each other,
such that one would find a human who is half donkey and half human, or a donkey that
is half bird and half donkey. But no [such creature] exists: rather, this is impossible. The
reason it does not exist is that all its benefits would be destroyed, and God does not destroy
anything [beneficial]. Moreover, the mixing of species with one another is impossible
because the species are connected to the individuals. Thus, it has been established that the
species do not mix with each other. Understand this!

(۲) و دیگر،166 آن چیز کی بهقوّت است و به فعل می‌آید، از خویشتن آن به فعل باز دهد کی درو بهقوّت
بود. اگر روا بَوَد کی آن167 نطفۀ خر، کی بهقوّت خر است، اسپ بَوَد، یا نطفۀ اسپ، کی بهقوّت اسپ است، خر
بَوَد، حکمتِ ایزد بهباطل بَوَد، و این نه موجود است. *فاعرفه*.

V. 3.2. Further: that which is in potentiality and comes to exist in actuality will by itself
bring into actuality that which was laid in it in potentiality. If it were permissible that the
sperm of the donkey, which is in potentiality a donkey, would become a horse, or that the
sperm of the horse, which is in potentiality a horse, would become a donkey, then God's
wisdom would be destroyed. But this never happens. Understand this!

(۳) امّا از پس جدا شدن از بدن، گُروهی ایدون پنداشتند[168] از نادانان، کی نوعی نوعی دیگر گردد، و ایشان آن گروه اند[169] کی به تناسخ[170] منسوب اَند[171]، کی روح مردم در جسمِ سگ و خر[172] شود و روح سگ و خر در جسم مردم آید. و از هذیان کی گویند این بَتَر از همه است، زیراک این بهتانی بزرگ است، کی ما همی بینیم کی نطفهٔ سگ[173] را صورتِ سگ درو[174] مقدور است[175] و از سگ بیرون آمد و در سگی شد و در آن نطفه موجود بود[176] هرچه آن[177] نطفه را به کار بایست[178] از پذیرفتنِ حسّ و حرکت و غیره، و میدیدیم آنک[179] همی اَفزود و همی کاست تا آنگه کی[180] در رَحِم او تمام شد. کدام وقتْ توهّم شاید کردن و جایز بود کی گوییم «روحِ مردم در آن سگ پیوست»؟ بدانک این حکمی باطل است. و چه فایده بَوَد درآمیختنِ[181] انواع با یکـدیگر[182] از پس جدا شدن از بدن؟ اگر ایدونک گویند: «ایزد عذاب کند گناهکاران را در اجسامِ بَهایم»، ایدون گوییم او را کی «اگر این دعوی درست است، در اجسامِ مردمان[183] کفایت است عذاب کردنْ گناهکاران[184] را، زیراک[185] اندر اجسامِ مردم بسیار اند[186] گندهتر و پلیدتر از سگان و گُرگان و خوکان.»

V. 3.3. Regarding the situation [of the soul] after [its] separation from body, a group of ignorant people assumed that one species [could] become another. These are the group to whom is ascribed the [doctrine of] metempsychosis (*tanāsokh*), which would have it that the soul (*rūḥ*) of a human can go into the body of a dog or a donkey, and the soul of a dog or a donkey into the body of a human. Of all the nonsense (*hadhayān*) they utter, this is the worst. It is a monstrous calumny! For we observe that the form of dog is preordained (*maqdūr*) in the sperm of a dog: it came out one dog and entered [the body of] another, and whatever is needed for the sperm to receive sense-perception, motion, and so forth, existed in that sperm. And we saw what constantly waxed and waned until it became complete in the womb. At which moment, then, could we imagine (*tavahhom*) or claim that the soul of a human being has joined that dog? You should know that this is a vain and empty judgment. Besides, what would be the benefit of mixing one species with another after [the soul's] separation from body? If they now say, "God punishes the sinners in the bodies of beasts", we reply to them, "if this claim is true, then being in human bodies is sufficient for the punishment of the sinners. Because among human bodies, there are many that are more corrupt and filthy than dogs, wolves, and swine".

(۴) و دیگر کی انواع را، هر یکی را[187]، بهری است و اندازهای[188] معلوم از اصلهای خویش کزان همی پدید آیند، و آن فلکها[189] کی اندر آن همی اثر کنند، کی هیچ یک را از آن جدا شدن نیست در صورتِ خویش، بلکی[190] هر نوعی را در آفرینش صورتی فلکی هست کی آن نوع را نگاه دارد. پس درست شد کی انواع با یکـدیگر[191] درنیامیزند، نه در ترکیب و نه بعد[192] از ترکیب. *فاعرفه.*

V. 3.4. Furthermore, for each species, there is a share and measure (*andāzeʰ*) determined by its principles, from which it manifests, and [by] those celestial spheres that have a constant effect upon them. None [of the species] may be separated in its form. Rather, there is for each species in creation a celestial form that preserves that species. Thus, it has been established that the species do not mix with each other, neither at composition, nor after composition. Understand this!

مقالتِ پنجم، جستارِ ششُم: در آنک[193] حرکتِ مُدوّر، یعنی جُنبِش گِرد، علّتِ حرکاتِ حیوانات است

(۱) بدانک هر چیز را[194] از فروغِ علّتی است، کی آن چیز پیش از آن اصولِ طبیعی پیدا نتواند[195] شد و نتواند پیدا شدن چیزی طبیعی بی علّت. و بدانک[196] حرکتِ حیوان[197] فروع طبیعت است و چاره نیست از آن، زیراک حیوان کی همی[198] پیدا شوند[199] فروع اند و نشاید بودن کی حرکتِ حیوانْ اصول بود[200]. پس چاره نیست کی آن را اصولی به کار باید، زیراک علّتها نسبت به معلولاتِ خویش بَرَد. و در عقل واجب آمد کی علّتِ حرکت هم حرکت بود، زیراک اگر ما ایدون گوییم کی علّتِ حرکت کمّیت است، این مخالف بود، از بهر آنک گفتیم «علّتِ مُشاکلِ معلول بود». و چون لازم آمد کی علّتِ حرکت هم حرکت بود از بهر مشاکلت، پس واجب آمد نظر کردن در اصولِ حرکت. نگاه کردیم تا کدام حرکت مشاکل بود حرکتِ حیوان[201] را از انواع حرکت. پس چون نگاه کردیم حرکات را، دیدیم کی قسمت برگرفت. قسمی حرکت مدوّر بود و قسمی حرکت مستقیم. پس چاره نیست کی علّتِ حرکتِ حیوان یا حرکتِ مستقیم بود یا حرکتِ مدوّر[202]. اگر ایدونک حرکتِ مستقیم علّتِ حرکت حیوان بود، لازم آید کی حرکتِ حیوان بر یک سوی بود، و این نه موجود است در حرکتِ حیوان، بل کی[203] حرکتِ حیوان به هر سو بود. پس چون حرکتِ حیوان به هر سو بود،[204] لازم شد کی حرکتِ مدوّر علّتِ حرکاتِ[205]حیوان بود. *فاعرفه*.

**Discourse V, Issue Six:** *That the circular motion, namely spherical motion, is the cause of animal motion*

V. 6.1. You should know that for everything among the offshoots (*forūʿ*), there is a cause, such that this [offshoot] cannot appear prior to those natural principles (*osūl-e ṭabīʿī*);

a natural thing cannot appear without a cause. And you should know that animal motion is an offshoot of nature, and is inevitable. For the animals that appear are themselves offshoots; therefore, animal motions cannot be principles. So it is inevitable that they are the work of certain principles, because the causes bear a relation (*nesbat . . . barad*) to their effects. Moreover, it becomes necessary by intellectual reasoning (*dar ʿaql*) that the cause of motion should itself be motion. For if were now to say that the cause of motion is quantity, relation, or substance, that would be contradictory, since we stated that the cause resembles (*moshākel*) its effect. Since, due to the resemblance (*moshākelat*), the cause of motion must itself be motion, we have to inquire into the principles of motion. We considered the different kinds of motion to [find out] which motion resembles that of the animals. When we considered the motions, we observed that they are of various types: one type is circular motion, another is rectilinear motion. Therefore, it cannot but be that the cause of the motion of animals is either rectilinear or circular motion. Now, if rectilinear motion were the cause of the motion of animals, it would follow that animal motion would [always] be in a single direction, but in fact this does not exist in the motion of animals. To the contrary, animals move in all directions. Since the motion of animals may be in any direction, it follows that circular motion is the cause of the motion of animals. Understand this!

(۲) و دیگر کی حرکتِ حیوان مانندۀ[206] حرکتِ مدوّر است، زیراک[207] حرکتِ مدوّر در جرمِ مدوّر ظاهر است[208] به کلّیت کاوّل[209] آن[210] جز آخر آن حن‌بود[211]. و دیگر، حیوان چون حرکت کند، حرکت ازو[212] پیدا شود در هر سوی، و نتوان اندیشیدن از میانِ حرکت اوّل و از میانِ حرکت آخر. *فاعرفه*[213].

V. 6.2. Further, the motion of animals is analogous (*mānande^h*) to circular motion, because the circular motion in a spherical body in general obviously begins and ends at the same point. Moreover, whenever an animal moves, it may manifest motion in any direction, and is no distinguishing between an initial point and an endpoint in the middle of motion (*az miyān-e ḥarakat*). Understand this!

(۳) و دیگر کی ظاهر است[214] و بهدلیل توان[215] کردن کی حرکتِ مدوّر علّتِ حرکاتِ حیوان است: آن است کی همی بینیم کی[216] حیوانْ حرکت کند آنگه کی خُرشید ظاهر شود، و چون خُرشید[217] به غُروب رسد کمتر حیوانات بیرون آیند از جایگاهِ خویش، و آنک[218] به شب بیرون آیند[219] چون خوک و گفتار و بوم و مار و گژدُم اند، و بیشترین چیزها کی در شب پدید آیند، بر اندازۀ شب باشند. ازین جهت[220] گفتیم حرکتِ مدوّر علّتِ حرکاتِ حیوان است. *فاعرفه*.

V. 6.3. Furthermore it is both evident, and something for which one can argue, that circular motion is the cause of animal motion: for we regularly observe that animals move around when the sun is up, and once the sun sets, animals seldom venture out of their lairs. The [nocturnal animals] that come out of their lairs during the night include the boar, the hyena, the owl, the snake, and the scorpion. Most creatures that appear during the night continue their motion [only] as long as night lasts. On this basis too, we say that circular motion is the cause of animal motion. Understand this!

مقالت ششم در خلقِ خامس[221] [=نبوّت و امامت]

**Discourse VI: On the Fifth Creation**

**Table A8.** The Seven Issues of the *Sixth Discourse* on the Prophethood and Shia Imamate.

| | |
|---|---|
| **VI. 1:** *How the prophethood of the Prophets is facilitated* | جستار اوّل: در چگونگی سَبُک گشتنِ پیغمبريِ پیغامبران |
| **VI. 2:** *On the dominion of the prophet over speech (sokhan) and the rhetoricians (ahl-e sokhan)* | اهلِ سخن را جستار دوم: در غلبه کردنِ پیغمبریِ سخن را و |
| **VI. 3:** *On why the later prophet confirms the veracity of the former prophet* | گوی داشتنِ پیغمبر پسینْ پیغمبر جستار سوّم: در علّتِ راست پیشین را |
| **VI. 4:** *On why the former prophet spreads the joyful news of the future prophet* | جستار چهارم: در علّتِ بشارتِ پیغمبر پیشین پیغمبر پسین را |

**Table A8.** *Cont.*

| | |
|---|---|
| **VI. 5:** *That the proof of God is not established by a single prophet* | جستار پنجم: در آنک به یک پیغمبر حجّتِ خدای بر پای نشود |
| **VI. 6:** *On the meaning of the attribution of "descent" to Jesus [alone] among all the prophets* | جستار ششم: در معنیِ نسبتِ فروآمدن به عیسی از میانِ همهٔ پیغمبران |
| **VI. 7:** *On the meaning of the attribution of Lord of Resurrection to the Mahdī* | جستار هفتم: در معنیِ نسبتِ خداوندِ قیامت به مهدی |

<div dir="rtl">

مقالت ششم، جستار اوّل: در چِگُونگيِ سَبُک گَشتنِ پیغمبريِ پیغامبران

(۱) هر چیزی<sup>222</sup> را از آفرینش مغزی هست کی چاره نیست کی آن چیز پیدا شود و منفعتِ خویش پیدا کند، آن منفعتها کی درو<sup>223</sup> نهاده است. و ما گفتیم کی<sup>224</sup> مغز حیوانْ مردم<sup>225</sup> ناطقِ زنده است و پیدا شد منفعتهای مردم، و آن پیشه‌های عجیب است کی مردم پیدا کرد به عقلِ خویش و به لطافتِ فطنت و به صَفوتِ هنر خویش. و همه کار<sup>226</sup> از آفرینش به مردم رسید، و از پس مردم در آفرینش صورتی پیدا نشد کی به شرف از مردم برگُذَشت<sup>227</sup>. واجب آمد کی مغز مردم در سخن بود، لیکن<sup>228</sup> همه سخنهای<sup>229</sup> مردم کی در یک دور برانند<sup>230</sup> چون شیر دوشیده بود کی در خیک همی زنند<sup>231</sup> تا از آنجا روغن گرد آید. و دیگر، آن سخنان مردمان کی در یک دور گرد آید و به لطافتِ آن پذیرفتَنْ یک تن را؛ و آن سَبُک گردد بر زبانِ او تا بدان پیغمبری بتواند پذیرفتن؛ و از بهر آن کی چنین بود آن سخن در دلِ قَومِ او جایگاه گیرد، کی آن سخن مغز آن سخنان است کی ایشان در مجلسِ خویش همی رانند<sup>232</sup>، تا گوش را شنیدن خوش آید از بهر آنک<sup>233</sup> آن رنگها است کی نزدیکِ ایشان بود. این است سَبُک گَشتنِ پیغمبریِ پیغامبران. *فاعرفه.*

</div>

**Discourse VI, Issue One:** *How the prophethood of the Prophets is facilitated*

VI. 1.1. There is for everything in creation a nucleus (*maghz*), from which things must appear, and which must manifest its benefit, [that is,] those benefits that are predestined in it. We have said that the nucleus of "animal" is the rational living human (*mardom-e nāṭiq-e zendeh*). The benefits of the human have been explained: they are the marvelous crafts (*pīshe^h-hā*) that human produced with his intellect, which are due to the subtlety of his innate instinct (*feṭnat*) excellence of his virtue (*honar*). The entire work (*kār*) of creation has culminated in the human, and after the human no form appeared in creation that surpasses the human in nobility. Now, it has been shown necessary that the nucleus of human lies in speech (*sokhan*). However, all the human speech articulated during a given age (*dawr*) is like milk that, after milking, is beaten in a vessel of skin, [and thus churned] such that butter is built up as a result. Furthermore, the human speeches that are built up during one age may, due their subtlety, be difficult to receive for just one human being; these would be facilitated for his tongue, so that he could thereby accept the [charge] of prophethood. As a result, that speech will settle in the hearts of his people. For this speech is the nucleus of those speeches that they articulate in their own gatherings, such that they appear delightful to the ear, on account of the fact that they are the tones that are appealing to [their ears]. This is how the prophethood of the prophets is facilitated. Understand this!

<div dir="rtl">

(۳) و سبک گَشتنِ پیغمبریِ پیغامبران را چگونگی‌ای<sup>234</sup> دیگر است، و آن پیدا شدنِ کار کردنِ عقل است، ازیرا کی<sup>235</sup> هر چیزی را از اوّلِ آفرینش تا به آخر کارکردنهایی است<sup>236</sup>، همچون پیدا کردنِ کارهای اُمّهات و اجرامِ علوی در مَوالید، و کار کردنِ قصدِ<sup>237</sup> نفس در نبات و حیوان و پیدا شدنِ خاصیتِ نفس در مردم از جهتِ<sup>238</sup> سخن. پس نمانَد چیزی را از آفرینش شوقی<sup>239</sup> به کار کردن مگر عقل را به خداوندی و بزرگواری، و درنخورَد<sup>240</sup> ذرّه‌ای از کردنِ کارها مگر غلبت کردن و بزرگواری پیدا کردن. و مِهتری و بزرگواری و غلبه کردن آن بود کی<sup>241</sup> غلبتِ او مر شریفتر چیزی را بود و آن مردم است. پس فراچیکد از عقل روشنایی‌ها؛ پس از<sup>242</sup> طبیعت و نفس شخصی بساختَند با تمامترِ اعتدالی و لطیفتر طبعی و کاملتر هیأتی<sup>243</sup>، تا قابلِ آن<sup>244</sup> تأییدِ عقل شد تا برکتِ آن به نفس و طبعت رسید و نیکی‌ها و برکت‌های<sup>245</sup> ایشان سبُک شد، تا از آن چیزهای بسیار پدید آمد. این است چگونگیِ سبک گَشتنِ پیغمبریِ پیغامبران. *فاعرفه.*

</div>

VI. 1.3. There is yet another way that prophethood is facilitated for prophets, namely the manifestation (*paydā shodan*) of the workings (*kār kardan*) of intellect. This is because, for everything from the beginning of creation to the end of its work, there is a purpose (*nahāyat*) [proper to it]. [We saw this when] explaining the functions of the elements (*ummahāt*) and celestial bodies in the [production of] composite generated things, or the intention (*qaṣd*) of soul in plants and animals, or the manifestation of the proper characteristic (*khāṣṣiyat*) of

soul in the human, with respect to speech (*sokhan*). Hence, nothing in creation has a desire to function, if not for the sake of intellect, due to its mastery and greatness (*khodāvandī-yo bozorgvārī*). Not even the tiniest grain would be appropriate to any function, were it not to make manifest the dominion and greatness [of the intellect]. And superiority, greatness, and dominion consists in [exercising] its dominion over the noblest of all things, this being the human. Thus lights flowed out from the intellect; then, an individual (*shakhṣ*) was fashioned from nature and soul, with the most complete harmony and subtlest natural constitution (*ṭabʿ*) and the most perfect disposition (*hayʾat*). As a result, [this individual] became receptive to the support of intellect (*taʾyīd-e ʿaql*), such that the blessings [of intellect] reached out to the soul and nature, and their goodness (*nīkī-hā*) and blessings were facilitated, giving rise to plentiful things. This is how the prophethood of the Prophets is facilitated. Understand this!

<div dir="rtl">

**مقالتِ ششم، جستارِ هفتم: در معنيِ نسبت خداوند قيامت به مهدى**

(۲) و چنین گفتند[246] کى چون مهدى بیاید گرگ و گوسفند به یک جا[247] آب خورند. پس معنيِ گرگ معنيِ ضدّ است کى[248] اوّلیاى خداى را ناهمواری و دشوارى[249] نمایند. و معنيِ گوسفند معنى[250] آن کس بود کى بدو ایمن باشند و از نیکيِ او اومید[251] دارند. پس ازین[252] معنى بود کى موافقت باشد میانِ ضدّ و ولى از قوّتِ خداوندِ قیامت. و معنيِ آب خوردن موافقت باشد میان ایشان در علم و حکمت و کشفِ حقایق.[253]

</div>

**Discourse VI, Issue Seven:** *On the meaning of the attribution of Lord of Resurrection to the Mahdī*

VI. 7.2. It is said that when the Mahdī comes, the wolves and the sheep will drink water from one and the same place. Here "wolf" means opponent (*ḍidd*): those who give trouble and difficulty to the friends of God (*awliyāʾ-e khodāy*). "Sheep" means the one in whom [the friends of God] find security, and in whose goodness they place their hope. It is in this sense, then, that there will be agreement between the opponent and the friend [of God] thanks to the power of the Lord of the Resurrection. "Drinking water" means agreement between them in knowledge, wisdom, and the unveiling of true realities.

<div dir="rtl">

**مقالتِ هفتم: در ذکرِ خلقِ ششم [= معاد، قیامت]**

</div>

**Discourse VII: On Mentioning the Sixth Creation**

**Table A9.** The Seven Issues of the Seventh Discourse on the the Resurrection and the Return to the One.

| | |
|---|---|
| **VII. 1:** *That resurrection is a peer of existence* | <div dir="rtl">جستارِ اوّل: در آنک برانگیختنْ قرینِ بودن است</div> |
| **VII. 2:** *That supposing multitude [of all living beings] to be resurrected is contrary to the truth* | <div dir="rtl">جستارِ دوّم: در آنک اندیشه در بسیار برانگیختن خلافِ حقّ است</div> |
| **VII. 3:** *That the knowledge of the resurrection is veiled from the soul* | <div dir="rtl">جستارِ سوّم: در آنک معرفتِ برانگیختنْ در حجاب است از نفس</div> |
| **VII. 4:** *That the resurrection is the discipline of the soul* | <div dir="rtl">جستارِ چهارم: در آنک برانگیختنْ ریاضتِ نفس است</div> |
| **VII. 5:** *That due to resurrection, the ill-fated may become well-fated, and the well-fated may become ill-fated* | <div dir="rtl">جستارِ پنجم: در آنک از بهر برانگیختنْ بدبخت نیک‌بخت گردد و نیک‌بخت بدبخت شود</div> |
| **VII. 6:** *On the length and shortness of the duration of the resurrection* | <div dir="rtl">جستارِ ششم: در درازی و کوتاهيِ مدّت برانگیختن</div> |
| **VII. 7:** *That regarding the resurrection, good deeds are of greatest benefit* | <div dir="rtl">جستارِ هفتم: در آنک کارهای نیک بزرگتر منفعتى‌ست برانگیختن را</div> |

## Notes

1. We express our gratitude to Hamed Arezaei for helping us access MS Mīnovī 2857, MS Adabiyāt 194jīm, MS Dāneshgāh-e Tehrān 8798, and MS. Malek 4055-2. Further, we extend our gratitude to Fateme Mehri for reviewing a previous draft of the edited passages and providing valuable insights.

2. For the sake of brevity, we omitted the differences in the footnotes to save space; however, we retained the divergences in words and expressions that include *kay/ki*[h]*/ke*[h], such as *ān-kay/ān-ke*[h] or *azīrāk/azīrāke*[h].

3. [کی] کر: که

4. [آفریدگار است] اد: آفریدهکار است / دت: آفریدهگارست

5. [یا] کر: حیا >،< مل: دت، اد، یا / مج: می‌ح، —— :می /

6. [نفس با خرد] مج، مل: خرد با نفس /

7. [کی] کر، می: که / مج، اد، دت: کی

8. [کی] کر، می، مج: که / اد، دت: کی

9. [این چُنان] مج: آن چنین / مل: اینچنین

10. [گه] دت: گاه، دت‌فوق: که

11. [را] مج، مل: ——

12. [ّس] مج، مل: ——

13. [جسم و روح] مج، مل: روح و جسم

14. [کی] اد، دت: که

15. [و] می:——/ مج، مل: +آن

16. [آنک] مج، مل: آنکه

17. [تمییز] مج‌ناخوانا، مل: تمیز

18. [بهر] مج، مل: بابت

19. [ملائکه] می: ملایکه / مج، مل، اد: ملئکه / دت: ملاءکه

20. Here nabāt, which often means "plant", is used in its original meaning of that which grows, so it includes both animals and plants.

21. [کی] مج، مل: ——

22. [به] مج، مل: با

23. [دو] مج، مل: بدو

24. [و] مج، مل: ——

25. [بهری ناطق و بهری غیر ناطق] مج، مل: ناطق و غیر ناطق

26. [و غیر ناطق] مل: ——

27. [پَراکنده‌ها] کر: پراکنده‌ها / می، اد: براکنده‌ها / مج، مل: پراکندکیها / دت: براکنده‌ها

28. [زیراک] مج، مل: زیرا کی

29. [مردم است] اد: مردمست

30. [در] دت: ء، دت‌فوق: در

31. [بهر است] کر، می، مج، مل: بهرست / اد، دت: بهر است

32. [زیراک] مج، مل: زیرا کی

33. [بهر] مج، مل: ——

34. [زیراک] مج، مل: زیرا کی / اد: زیرا

35. [حسّی و ... زیراک زندگانی] دت: ——، دت‌ح: حسّی و ... زیراک زندگانی / [زندگانی] مج، مل: نفس زندگانی

36. [آنک] مج، مل: آنکی

37. [اندر] مج، مل: در

38. [طبیعت است] مج: بطبیعتست / مل: بطبیعت

39. [ازین] مج: از این

40. [جهت] اد: جهة

41. Corbin and Landolt translate as "quintessence". Corbin explains the meaning of the words lobb and maghz as follows: "The heart, marrow, nucleus, or innermost substance of a thing".

42. [یافت] مج، مل: گرفت

43. [پذیرفتن] دت: بردن، دت‌فوق: پذیرفتن

44. [موالید] اد: مولید

45. [فایده‌ای] کر: فایده‌ئی / می: فایده□ / مج: فایده، دت / دا، مل: فایده

46. [نبات] مج، مل: +و

47. [فایده‌ای] کر: فایده‌ئی / می: فایده□ / مج: فایده، دت / دا، مل: فایده

48. [گیاه‌ها و میوه‌ها] کر، می: گیاها و میوها / می‌ح: کذا / مج، دت: گیاها و میوه‌ها / اد: گیاها و میوه‌ها

49. [دادن] دت: یافتن، دت‌فوق: دادن

50. [پذیرفتن] دت: بردن، دت‌فوق: پذیرفتن

51. [را] مج: ——

52. [پذیرفتن] مج‌ح، ملح: +و دادن با نبات «احتمال دارد این عبارت در اصل ساقط شده»

53. [دادن] مج، مل: پذیرفتن و دادن

54. ‏[قبل] مل: قبیل
55. ‏[را] مج: از بهر حرکت
56. ‏[است] مج، مل: ——
57. ‏[زندگانی] دت: زندگی، دتفوق: زندکانی
58. ‏[طبیعی] مج، مل: +ست
59. ‏[کسب] اد: سبب
60. ‏[فاعرفه] اد: ——
61. ‏[در] کر، می، اد، دت:——/ مج، مل: در
62. ‏[از آن] کر: از / می، مج، اد، دت: از آن
63. ‏[نبینی] مج، مل: و نیست
64. ‏[یکدیگر] کر، می: یک دیگر
65. ‏[چون دوست داشتن لونهای نیکو و رویهای نیکو بود بصر را؛ و غلبتِ بصر] مج: ——
66. ‏[نفس] مجصح، مل: +نگاه
67. ‏[شمّ] دت: نسیم ← دتفوق: شمّ
68. ‏[خوش] کر، می، دت، دا:——/ مج، مل: +و
69. ‏[بگوارانیدن] می: نکوارانیدن
70. ‏[حَرب] اد: +کردن
71. ‏[آن] مج، اد، مل: از / دت: ——
72. ‏[از نفس] مج، مل: ان نفس را
73. ‏[بیم] دت: ——، دتفوق: بیم
74. ‏[زیراک] دا: زیرک
75. ‏[حیوة] دت: حیات / اد: حیوات
76. ‏[صنعت] اد: طنعت
77. ‏[آن] کر: این
78. ‏[رنگهای آن] مل: رنگها وان
79. ‏[مُشاکل است] کر: مشاکلست / اصل (کر)، می، مج، اد، دت، مل: مشاکلت
80. ‏[روحانی] مج: جسمانی، مجفوق: روحانی / ملفوق: جسمانی
81. ‏[نبینی] مج، مل: ببینی / [مشاکلت او ... نبینی، ... و نه معدنی،] کر: مشاکل است او به جوهر روحانی. نبینی ... و نه معدنی؟
82. ‏[ولیکن] مج: ولکن
83. ‏[آنچ] مج، مل: آنچه
84. ‏[آنک] مل: انکه
85. ‏[نگاه داشته] مل: نگاهداشته
86. ‏[کی] مل: که
87. ‏[نگاهداشت] مل: نگاهداشت
88. ‏[نگاه داشتن] مج، مل: نگاهداشت
89. ‏[زَبَر] می: زیر، میح: زبر / مج: زیر / مل: در زیر
90. ‏[زیراک] مل: زیرا که
91. ‏[گشتن] اد: شدن
92. ‏[دو] میناخوانا: رو / دتفوق: ی
93. ‏[کی] مل: که
94. ‏[نگاه دارد] مل: نگاهدارد
95. ‏[نگاه داشته] مل: نگاهداشته
96. ‏[واجب است] اد: وجبست
97. ‏[جهت || جهت || جهت] اد، دت: جهة || جهة || جهة
98. ‏[بلک] مل: بلکه
99. ‏[نگاه داشته] مل: نگاهداشته
100. ‏[بسیار است و آن یک نوع است؛ مثل] کر: بسیارست، و آن یک نوع است مثل
101. ‏[ایدونک] مل: ایدونکه
102. ‏[بر] دت: در، دتفوق: بر
103. ‏[آنچ] مج: آنچه
104. ‏[آنچ] کر: آنچه / می: آنچ / دت: آنک، دتفوق: نچ
105. ‏[امّا] دت: و امّا
106. ‏[آنچ] مج: آنچه
107. ‏[آنچ] مج: آنچه
108. ‏[آنک] مج: آنکی
109. ‏[نه] مج، اد، دت: ——
110. ‏[اثر هم از آن اصل] مل: ازهم از آن اصل
111. ‏[در آفرینش بر فصلِ اوّل اند، از بهرِ آنک این اثر از آن شعاعاتِ صورتهایِ فلکی پذیرند] مل: ——
112. ‏[نگاه دارنده] مل: نگاهدارنده
113. ‏[چیز کی || آنک || زیراک || کی در || آنک] مل: چیزیکه || زیرا که || آنکه || که || در ||
114. ‏[را] کر، می، مج، اد، مل: ——
115. ‏[پیش ... پیش] می، اد، دت: بیش ... بیش
116. ‏[خواستیم] مج: خواستم
117. ‏[کی] مج:——/ [کی] مل: ——

118. ‏[درین] مج، مل: در این‏
119. ‏[آنچ] مج، مل: آنچه‏
120. ‏[همه چیزها] مج: + همه چیزها‏
121. ‏[پیش] می، اد، دت: بیش‏
122. ‏[باید] مج، مل: آید‏
123. ‏[همچنانک] مج: هم چنانک‏
124. ‏[مدبّر] مل: بر‏
125. ‏[کزان] کر: کز آن / می، مج، اد، مل: کزان‏
126. ‏[و این] دت: و ان ← و این‏
127. ‏[باید] اد: بایست‏
128. ‏[پیش] می، دت، اد: بیش‏
129. ‏[کی] مل: ——‏
130. ‏[از] مج: ——‏
131. ‏[دربودن] کر: و بودن / اصل (کر): ودبوذن / می، اد، دت: وبودن / مج، مل: و دربودن‏
132. ‏[آن] اد: ان آن‏
133. ‏[آمدن] می: آمذان، میح: کذا فی الأصل / مج: آمدان / اد، دت: آمذ آن / مل: آمد آن‏
134. ‏[همچنانک || زیراک || ایدونک کی || ایدونک] مل: [همچنانکه || زیرا که || گوید || ایدونک که || ایدونکه]‏
135. ‏بر> بودن / می: بودن / مج، اد، مل: ببودن / دت: ——<به بودن] کر: ‏
136. ‏[پیشتر || پیشتر|| پیشتر || پیشتر || پیشین] می، اد: بیشتر || بیشتر || بیشتر || بیشتر || بیشین‏
137. ‏[بکردند] اد: نکردند‏
138. ‏[فلک] مل: انکه‏
139. ‏[بیشتر || پیشتر] می، اد: بیشتر || بیشتر / کر، مج: بیشتر || پیشتر‏
140. ‏[شد] اد: باشد‏
141. ‏[فراپیش] دت: آفرینش ← فراپیش‏
142. ‏[قوت || قوت || قوتها || قوت || قوت || قوت] اد: قوّت || قوّت || قوّتها || قوّت || قوّت / دت: قوّت || قوّت || قوّتها || قوّت || قوّت || قوّت‏
143. ‏[پیشتر] می، مج، اد، دت: بیشتر /‏
144. ‏[زیراک] مج: زیرا کی / مل: [زیرا که]‏
145. ‏[کند] مل: کنند‏
146. ‏[و گرد همی کند] مج: ——‏
147. ‏[باز] مج: باو‏
148. ‏[آن چیز است] دت: انست ← آنچیزست‏
149. ‏[پیشهها || پیشهها] کر: می: بیشها / مج، مل: پیشها / اد، دت: بیشهها‏
150. ‏[علمی است] مج، مل: علمست‏
151. ‏[از] کر: از / می، مج، اد، مل: ——‏
152. ‏[زیراک || آنک] مل: [زیراکه || آنکه]‏
153. ‏[است] مج، مل: ——‏
154. ‏[بردن] دت: بوذن ← بردن / [شغل بردن] مل: مشغول بودن‏
155. ‏[آنچ] مج، مل: آنچه‏
156. ‏[ننهادند] می، اد: بنهادند / مج، دت: نهادند‏
157. ‏[درست شد ... نبات بود.] اد: پس درست شذ کی قول آنان کی کفتند کی حیوان از بس نبات بدیذار آمد ناروا بوذ و ایشان را بر این قول برهانی نباشذ تا ببر هان ثابت کنند بس این درست بوذ کی حیوان نه از بس نبات بوذ‏
158. ‏[یکدیگر] مج: یکدیگر / مل: یکدیگر‏
159. ‏[زیراک] مج: زیرا کی / مل: زیرا که‏
160. ‏[درآمیختی] مل: درآمیختن‏
161. ‏دیگر] مل: ——[جایز بودی ... با یک‏
162. ‏[مردمی] دت: مردم، دتفوق: می‏
163. ‏[شد] مج: شده‏
164. ‏دیگر ... یکدیگر ... یکدیگر] مج، اد، دت، مل: یکدیگر[یکدیگر ... یک‏
165. ‏[درنیامیزند] دت: نیامزند، دتفوق: در‏
166. ‏[دیگر ] اد: +کی‏
167. ‏[آن] دت: ——، دتفوق: آن‏
168. ‏[ینداشتند] اد: بیذاشذند‏
169. ‏[گروه اند] اد، مج: گروهند‏
170. ‏[به تناسخ] اصل (کر): نتناسخ‏
171. ‏[منسوب اند] مج: منسوبند / اد: +کی گویند‏
172. ‏[سگ و خر] دت: خروسک‏
173. ‏[سگ] مل: یک‏
174. ‏[درو] اد: در او‏
175. ‏[صورتِ سگ درو مقدور است] مج، مل: جز صورت سگ درو مقدور نیست‏
176. ‏[بود] دت: شد، دتفوق: بود‏
177. ‏[هرچه آن] مج، مل: هرانچه‏
178. ‏[بایست] اد: باید / مل: بست‏
179. ‏[آنک] مج، مل: ——‏
180. ‏[آنگه کی] مل: انکه‏

181. ‏[درآمیختن] کر: در آمیختن
182. ‏[با یکدیگر] مج، مل: بیکدیگر / اد، دت: با یکدیگر
183. ‏[مردمان] دت: مردم، دتۇفوق: مان /
184. ‏[گناهکاران ... گناهکاران] مج: کناه کاران ... کناهکاران / می، اد: کناه کاران ... کناه کاران
185. ‏[زیراک] مل: [زیرا که]
186. ‏[بسیار اند] می، مج، اد، دت، مل: بسیارند
187. ‏[را] مج، اد، مل: ——
188. ‏[اندازه‌ای] مج، مل: اندازه
189. ‏<ست>[و آن فلک‌ها] می، اد، دت، مل: وآن فلکها / کر: و آن فلکها
190. ‏[بلکی] مل: [بل که]
191. ‏[یکدیگر] اد، دت: یکدیگر
192. ‏[بعد] مج، مل: پس
193. ‏[آنک] مل: آن که
194. ‏[چیز را] مل: چیزیرا
195. ‏[نتواند] مل: تواند
196. ‏[بدانک || بدانک] مل: بدانکه || بدانکه
197. ‏[حیوان] مج، مل: +از
198. ‏[همی] مج، مل: ——
199. ‏[شود] مج، مل: شوند
200. ‏[بود] مل: بوند
201. ‏[حیوان را] می، مج، اد، دت، مل: حیوان را / کر: حیوانانرا
202. ‏[مستقیم ... مدوّر] اد: مدوّر ... مستقیم
203. ‏[آنک || ایدونک || بل کی] مل: [انکه || ایدون که || بل که]
204. ‏[پس چون حرکتِ حیوان به هر سو بود،] مج، مل: ——
205. ‏[حرکات] مج، مل: حرکت
206. ‏[مانندهٔ] مج، مل: مانند
207. ‏[زیراک] دت: زیرا که
208. ‏[ظاهر است] اصل (می): ظاست
209. ‏[کاوّل] مج: کی اوّل / [زیراک || کاوّل] مل: زیرا که || که اوّل
210. ‏[آن] مل: ——
211. ‏[حن>بود] کر، می، مج: بوذ / اد، دت: بود

    If we read joz (particle or part) instead of joz (but, different from)—assuming that the Hamza is dropped from the end of joz— then we get the same sense without amending bovad to nabovad: "its initial point is its last part" (ke-avval-e ān joz-e akhar-e ān bovad).

212. ‏[ازو] مل: آزاد
213. If read as "از میانِ حرکتِ اوّل و از میان حرکتِ آخر" (az miyān-e ḥarakat-e avval va az miyān-e ḥarakat-e āḫar), namely with two ezāfas in each nominative chain, then we get, "one cannot distinguish [think of] the middle of the initial motion from the middle of the final motion". However, if we do not read ḥarakat as a further second term of two ezāfas with avval and āḫar as their third terms, namely if we read both occurrences of ḥarakat with sukūn (az miyān-e ḥarakat avval va az miyān-e ḥarakat āḫar), then: "one cannot distinguish between [think of] an initial point in the middle motion and a final point in the middle motion" (which stands in translation).

214. ‏[ظاهر است] اصل (می): ظاست
215. ‏[توان] اد: +ثابت
216. ‏[کی] مج، مل: ——
217. ‏[خُرشید ... خُرشید] مج، مل: خورشید ... خورشید
218. ‏[آنک] مج: انکی / مل: انکه
219. ‏[آیند] مج، مل: آید
220. ‏[جهت] اد: جهة / [ازین جهت] مل: اراینجهت
221. ‏[خامس] مل: خاص
222. ‏[چیزی] مج، مل: چیز
223. ‏[درو] مج: در او
224. ‏[کی] مج، مل: یکی
225. ‏[مردم] مج:——/ [کی مغز حیوانْ مردم ناطق زنده است] مج: یکی مغز حیوان ناطق زنده است
226. ‏[کار] مل: کارها
227. ‏[برگُذَشت] مج، مل: بگذشت
228. ‏[لیکن] مج، مل: لکن
229. ‏[سخن‌های] دت: سخن‌ها
230. ‏[برانند] مج، مل: برایند
231. ‏[زنند] مل: نزنند
232. ‏[رانند] مل: دانند
233. ‏[آن کی || آنک] مل: [انکه || انکه]
234. ‏[چگونگیای] کر: چگونگیی / می: چگونگئی / مج، اد، دت، مل: چگونگی
235. ‏[ازیرا کی] مج: زیراکی / مل: زیرا که

236. [ کارکردن‌هایی است] مج: کارکردنهایییست / کر، می، دت: کارکرد نهایتیست / اد: کارکرد نهایتی است / مل: کارکردن نیست
237. [قصد] اد: ——
238. [جهت] اد: جهة
239. [شوقی] مجح، مل: +نباشد
240. [درنخورَد] مجفوق-ناخوانا: بان / مل: +بان ا
241. [کی]مل: که
242. [از] مل: ——
243. [هیأتی] اد: هینتی
244. [آن] مج، اد، مل: از
245. [برکت‌های] مل: برکتها[کی]مل: که
246. [و چنین گفتند] مج:——/ اد: و دیکر کفتند
247. [جا] اد: جای
248. [کی] اد: یعنی
249. [دشواری] اد: درشتی
250. [معنی] اد: ——
251. [اومید] مج: امید
252. [ازین] مج، اد: از این
253. [مقالتِ ششم، جستار هفتم ... و حکمت و کشفِ حقایق.] مل: ——

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
