# Peer review of "The Necessity and Goodness of Animals in Sijistānī’s Kashf Al-Maḥjūb"

_philosophies, doi:10.3390/philosophies9030072_

Round 1
Reviewer 1 Report
Comments and Suggestions for Authors
The article is perfectly complete and enjoying to read. However, I would like, although it is not necessary, to raise concern on the following issues that emerge from within the paper's treatment, as thoughts that the author might wish to address and comment upon.
It would be interesting, and perhaps worth it, to discuss further the natural and logical determinism that seems to weaken the value of the divine will.
On line 418, where it is mentioned that God wills, one wonders what kind of freedom is present, if forms and species and God's substance is taken for granted and interconnected.
In the sentence of lines 409-410 there seems a verb to be missing?
Reviewer 2 Report
Comments and Suggestions for Authors
The article under review focuses on the philosophical and theological aspects of the significance of animals within the context of the thought of Abū Yaʿqūb al-Sijistānī, a representative of the Ismaili stream in Islamic philosophy. The text explores how animals are encompassed within the cosmological concept of emanation, characteristic of Neoplatonism and its adaptation in Islamic philosophy, emphasising the necessity and beneficence of animals as an integral part of the cosmic order. The vocabulary employed in the article is precise and appropriate to the academic nature of the work. The author demonstrates a profound understanding of the subject and the ability to communicate it effectively. The tone is neutral and objective, which facilitates a credible presentation of arguments. However, at places, there could be more explanations of complex concepts for readers unfamiliar with the subject matter.
The article is well-organised and divided into sections, which aids in understanding and following the argumentation. Each chapter focuses on a specific aspect of the issue under analysis, contributing to a logical and coherent construction of the text. Nonetheless, the introduction could include a more detailed review of the literature to better contextualise the study within existing works on the topic.
The work offers interesting insights into the role and significance of animals in Ismaili philosophy, which constitutes an important contribution to the understanding of this intellectual tradition. Although the topic is not entirely new, the approach and interpretation are quite original. However, the author could further emphasise how their work stands out against previous research in this field.
The article constitutes a valuable contribution to research on Islamic philosophy, offering a thoughtful analysis of the role of animals in Sijistānī's thought. The author is recommended to make minor revisions, such as a more detailed introduction to the research context and an expansion of the sections concerning the novelty of the research. Additionally, adding comparisons with other philosophical traditions could enrich the discussion and highlight Sijistānī's unique approach to the subject of animals.
The appendix to the article includes a detailed description of the manuscripts and the methodology of editing the source texts used in the work. This is a key element for understanding the research process and confirms the diligence with which the author approaches the analysis of Sijistānī's texts. The presentation of various manuscripts, their origins, as well as an explanation of the critical editorial apparatus and approach to transliteration, attest to a high level of scholarly craftsmanship.
Author Response
Our thanks for this positive review!
Reviewer 3 Report
Comments and Suggestions for Authors
This is a very detailed, original, and well-learned study based on the Neoplatonic notion of 'emanation'. It delves into Plotinus' thought and considers approaches based on Persian and Arabic philosophers who are relatively unknown in Western academic contexts. Moreover, the study employs perspectives that are yet to receive the attention they deserve. The arguments are extremely thorough and each sentence is very well cited.
This is a solid academic piece of work that I would like to share with my undergraduate and postgraduate students (although I think it might sound somewhat difficult for undergraduates in general). For this reason, and considering the impressive and thorough analysis offered in each section and paragraph, I am pleased to consider this study for publication without recommending any revision so far.
Author Response
Many thanks for the positive review!